# *Chlamydia* evasion of neutrophil host defense results in NLRP3 dependent myeloid-mediated sterile inflammation through the purinergic P2X7 receptor

Chunfu Yang[1], Lei Lei[1], John W. Marshall Collins[1], Michael Briones[1], Li Ma[1], Gail L. Sturdevant[2], Hua Su[1], Anuj K. Kashyap[1], David Dorward[3], Kevin W. Bock [4], Ian N. Moore[4], Christine Bonner[5], Chih-Yu Chen[5], Craig A. Martens[6], Stacy Ricklefs[6], Masahiro Yamamoto [7], Kiyoshi Takeda[8], Yoichiro Iwakura [9], Grant McClarty[10] & Harlan D. Caldwell [1✉]

*Chlamydia trachomatis* infection causes severe inflammatory disease resulting in blindness and infertility. The pathophysiology of these diseases remains elusive but myeloid cell-associated inflammation has been implicated. Here we show NLRP3 inflammasome activation is essential for driving a macrophage-associated endometritis resulting in infertility by using a female mouse genital tract chlamydial infection model. We find the chlamydial parasitophorous vacuole protein CT135 triggers NLRP3 inflammasome activation via TLR2/MyD88 signaling as a pathogenic strategy to evade neutrophil host defense. Paradoxically, a consequence of CT135 mediated neutrophil killing results in a submucosal macrophage-associated endometritis driven by ATP/P2X7R induced NLRP3 inflammasome activation. Importantly, macrophage-associated immunopathology occurs independent of macrophage infection. We show chlamydial infection of neutrophils and epithelial cells produce elevated levels of extracellular ATP. We propose this source of ATP serves as a DAMP to activate submucosal macrophage NLRP3 inflammasome that drive damaging immunopathology. These findings offer a paradigm of sterile inflammation in infectious disease pathogenesis.

[1] Laboratory of Clinical Immunology and Microbiology, National Institute of Allergy and Infectious Diseases, National Institutes of Health, Bethesda, MD, USA. [2] Laboratory of Virology, Rocky Mountain Laboratories, National Institute of Allergy and Infectious Diseases, National Institutes of Health, Hamilton, MT, USA. [3] Research Technology Branch, Rocky Mountain Laboratories, National Institute of Allergy and Infectious Diseases, National Institutes of Health, Hamilton, MT, USA. [4] Infectious Disease Pathogenesis Section, Comparative Medicine Branch, National Institutes of Allergy and Infectious Diseases, National Institutes of Health, Bethesda, MD, USA. [5] National Microbiology Laboratory, Public Health Agency of Canada, Winnipeg, Manitoba, Canada. [6] Genomics Unit, Research Technology Branch, Rocky Mountain Laboratories, National Institute of Allergy and Infectious Diseases, National Institutes of Health, Hamilton, MT, USA. [7] Department of Immunoparasitology, Research Institute for Microbial Diseases (RIMD), Osaka University, Osaka, Japan. [8] Laboratory of Mucosal Immunology, WPI Immunology Frontier Research Center, Osaka University, Osaka, Japan. [9] Institute for Biomedical Sciences, Tokyo University of Science, Tokyo, Japan. [10] Department of Medical Microbiology, University of Manitoba, Winnipeg, Manitoba, Canada. ✉email: hcaldwell@niaid.nih.gov

**C**hlamydia trachomatis is one of the most common human bacterial pathogens that cause blinding trachoma and sexually transmitted infection (STI). Trachoma is a disease of developing countries and 1.9 million people are blinded or severely visually impaired[1]. C. trachomatis is the most common cause of bacterial STI with 1.7 million reported cases of infection in the USA alone[2].

A sequela of female infection is a pelvic inflammatory disease (PID) that may result in tubal factor infertility and ectopic pregnancy[3]. The pathophysiology of trachoma and PID is not understood but episodes of reinfection[4,5] or chronic infection[6,7] that drive damaging inflammatory immunopathology are important. Innate immunity is implicated in the immunopathogenesis of trachoma and PID[8]. Neutrophil aggregation was

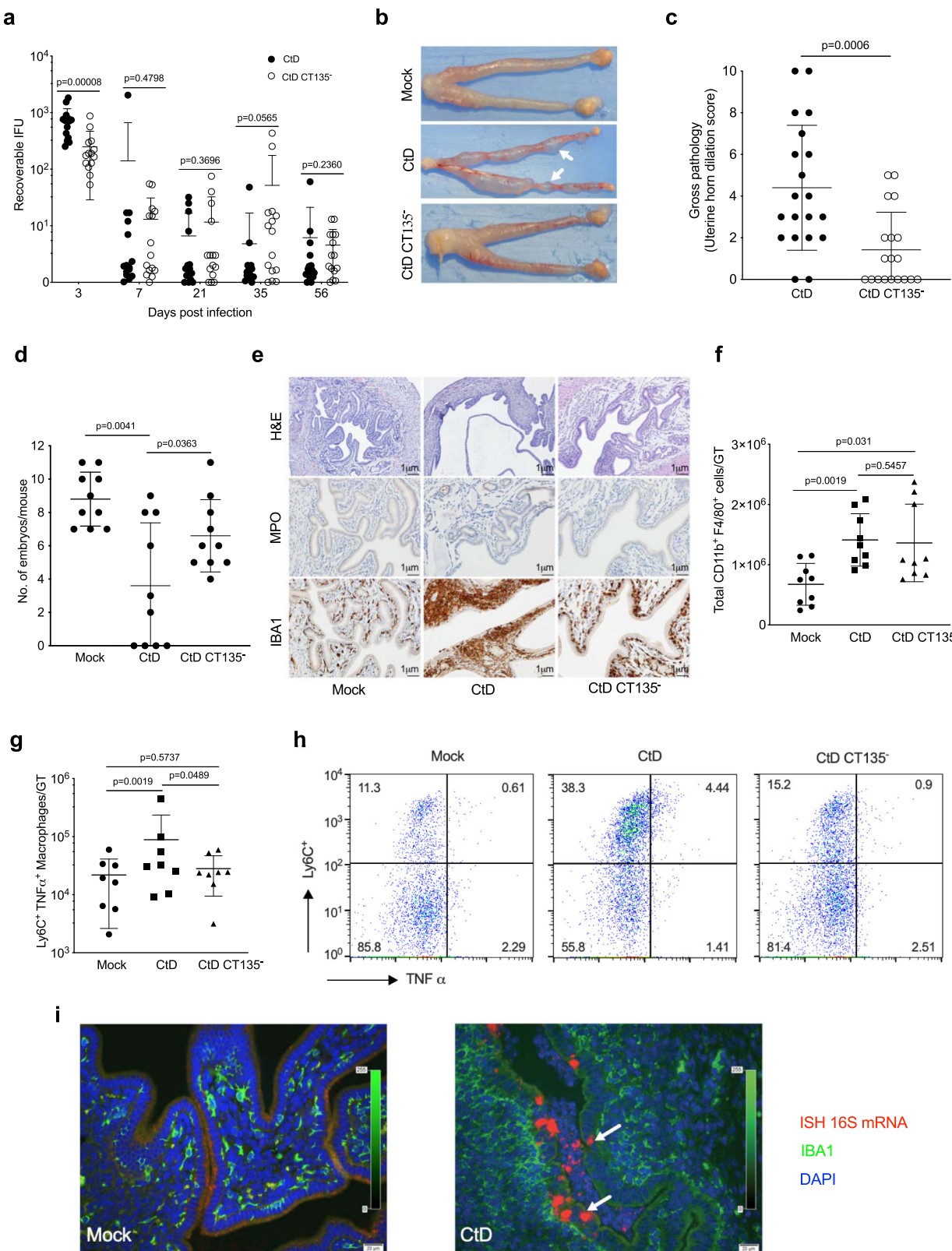

**Fig. 1 Infection and genital tract pathology of CtD and CtD CT135⁻ infected C57BL/6 female mice. a** Infection kinetics. CtD infected mice exhibit significantly higher burden at day 3 pi ($n = 15$ mice per chlamydial strain, two-tailed Mann–Whitney test). **b** Gross genital tract pathology at day 56 pi. White arrows denote uterine horn dilation with associated protrusions extending through the myometrium. **c** Quantification of gross uterine horn pathology. Clinical disease severity of CtD and CtD CT135⁻ infected mice was scored (0–10) by counting the total number of protrusions on both uterine horns from individual mice. ($n = 19$ mice per chlamydial strain, two-tailed Mann–Whitney test). **d** Infertility of infected mice 6 weeks pi. ($n = 10$ mice per group, two-tailed Welch's test). **e** Histopathology and immunohistochemistry. Histopathology in CtD infected mice at day 56 pi was characterized by severe endometrial gland dilation, stromal atrophy, and leukocyte infiltration comprised primarily of neutrophils and macrophages, whereas these histological changes in CtD CT135⁻ infected mice were minimal to absent. IBA1 staining showed submucosal macrophage infiltration in CtD and CtD CT135⁻ infected mice. **f** Flow cytometry of total CD11b⁺ F4/80⁺ macrophages in the whole GT of CtD and CtD CT135⁻ infected mice at day 56 pi ($n = 8$ mice per bacterial strain, two-tailed Mann–Whitney test). **g** Flow cytometry of total CD11b⁺ F4/80⁺ Ly6C⁺ TNF-α⁺ macrophages in the whole GT of CtD and CtD CT135⁻ infected mice at day56 pi ($n = 8$ mice per group, one-tailed student's $t$ test). **h** Flow cytometry CD11b⁺ F4/80⁺ Ly6C⁺ TNF-α⁺ macrophages of representative mice in panel (**g**). **i** Fluorescence immunohistochemical staining and in situ hybridization of chlamydial 16S mRNA at day 56 pi. Chlamydial 16S mRNA detection was found in uterine epithelial cells and luminal neutrophils in CtD infected mice (white arrows). Importantly, no chlamydial 16S mRNA was detected in IBA1-positive submucosal macrophages. **a, c, d, f, g** Data are pooled from two independent experiments and are shown as the mean ± SD. **e, i** Images are shown are representative of $n = 3$ mice per group.

observed in the endometrial histopathology of chlamydial PID patients[9] implicating a role of neutrophils in disease pathogenesis.

Transcriptional profiling of trachoma conjunctival samples showed transcriptional networks connected to the innate immune response[10,11]. Biomarker studies of women with chlamydial endometritis found increased expression levels of myeloid cell-associated inflammation[12]. Notably, polymorphisms in TNF-α and NLRP3 inflammasome pathway genes were linked to disease severity of trachoma and PID[13–16] suggesting a role of innate immunity in immunopathology. Paradoxically, chlamydial inflammatory diseases commonly exist with an absence or paucity of detectable infectious chlamydiae[5] with accompanying severe inflammation; this enigma is not understood.

*C. trachomatis* is an obligate intracellular bacterium distinguished by unique developmental biology that consists of two distinct development types: a small infectious elementary body (EB) and a larger non-infectious reticulate body (RB). Following attachment and entry, EBs undergo primary differentiation into RBs within a protective parasitophorous vacuole termed the inclusion. RBs closely associate with the inclusion membrane (IM) and synthesize a large family of structurally related proteins that are inserted into the IM termed IM proteins (Incs)[17,18]. A role for Incs in chlamydial pathogenesis is largely under-appreciated; only a limited number have been characterized functionally and are involved in the establishment and maintenance of the IM, acquisition of nutrients from the host, and counteracting host type I interferon response[19]. *C. trachomatis* CT135 is an IM protein[18] and a known virulence factor[20] but its function is unknown. There is a strong selection among human clinical isolates to maintain an intact CT135 gene[20], yet the gene is rapidly disrupted following in vitro propagation[21] suggesting CT135 as an important virulence factor in the pathogenesis of human infection.

Damage-associated molecular patterns (DAMPs) play a central role in the pathogenesis of human inflammatory diseases such as autoimmune diseases and cancer[22]. DAMPs are released upon cellular stress or tissue injury that triggers macrophage activation. Adenosine triphosphate (ATP) binds the P2X purinoceptor 7 (P2X7R) activating the NLRP3 inflammasome[23,24]. P2X7R is associated with increased susceptibility to bacterial and parasitic infections[25]. P2X7R aggravates tuberculosis disease progression and associated tissue damage[26,27]. Importantly, in that study P2X7R-mediated damage was restricted to infected macrophages at the primary infection site. Whether P2X7R functions in pathogen-induced inflammation at tissue locations distinct from the primary infection site has not been reported.

Here, we show that CT135 is essential for NLRP3 inflammasome activation in neutrophils through a TLR2/MyD88 signaling

pathway to evade neutrophil host defense. Conjointly, this pathogenic strategy drives submucosal macrophage-associated endometritis through ATP-induced NLRP3 inflammasome activation independent of chlamydial infection. These findings provide important advances in understanding the role of CT135 in the pathogenesis of chronic chlamydial infection and offer a model of sterile inflammation in infectious disease immunopathogenesis.

## Results

**Infectivity and pathology of female mice challenged with *C. trachomatis* CtD and CtD CT135⁻ organisms.** Female C57BL/6 mice were infected transcervical with $1 \times 10^5$ IFU of CtD and CtD CT135⁻. Infection was monitored by culturing cervicovaginal swabs on monolayers of McCoy cells and quantitating rIFU. CtD infected mice exhibited significantly greater infectious loads at day 3 post infection (pi) but thereafter no significant differences between strains were found throughout the 56-day pi culture period (Fig. 1a). The failure of both strains to spontaneously eradicate infection is because they possess the chlamydial plasmid, a virulence factor essential for establishing persistent infection[28]. The only significant difference in infection kinetics between strains was at day 3 pi, therefore we thought CT135 might function in colonization or early infection interactions with genital tract (GT) epithelial cells. The strains had similar ID₅₀ values (Supplementary Fig. 1) suggesting colonization efficacy was not the reason for the differences in early post infection chlamydial loads between strains. Allelic genetic complementation is not available in chlamydiae and anhydrotetracycline-induced plasmid-based complementation of CtD Ct135⁻:: bla CT135 is highly toxic beyond early time points post infection (Supplementary Fig. 2).

We next evaluated gross uterine horn pathology in CtD and CtD CT135⁻ infected mice. No gross pathological changes in GT tissues were observed in either CtD or CtD CT135⁻ infected animals at day 28 pi. However, CtD infected mice exhibited severe gross uterine horn pathology at day 56 pi that was remarkably reduced in CtD CT135⁻ infected mice (Fig. 1b, c). Uterine horn gross pathology was characterized by dilated glandular ducts that penetrated the myometrium resulting in visible protrusions[29]. Uterine horn pathology caused by CtD infection resulted in a significant increase in infertility (Fig. 1d). Histopathologically, CtD infection caused extensive endometrial gland dilation, stromal atrophy, and leukocyte infiltration comprised primarily of luminal neutrophils and submucosal macrophages (Fig. 1e). These histopathological changes were absent or minimal in CtD CT135⁻ infected animals. To ascertain if the total number of macrophage in the GT tissues of infected

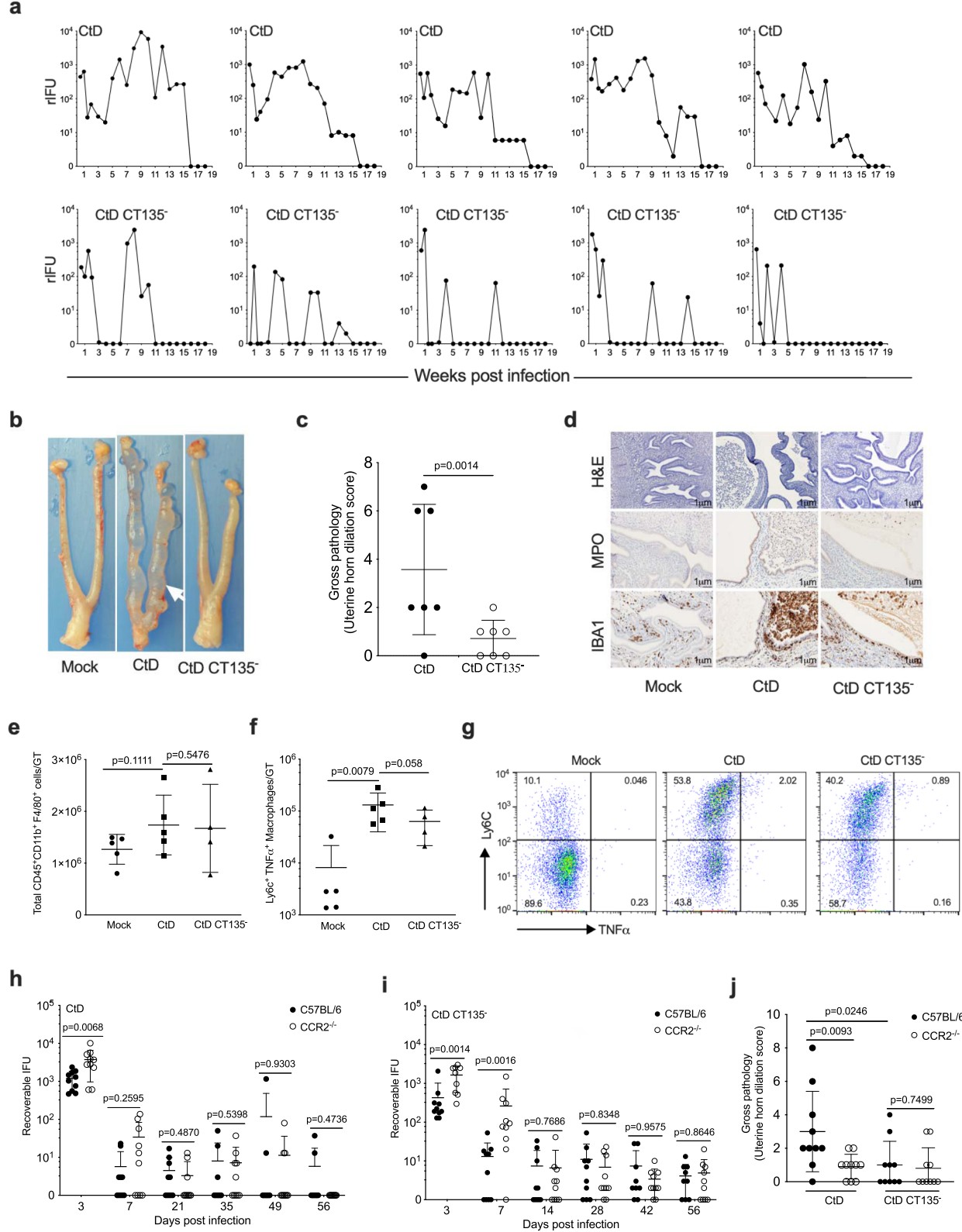

mice differed between CtD and CT135−, total leukocytes were isolated from the entire GT and CD11b+F4/80+ macrophages were quantitated by flow cytometry. We found total CD11b+F4/80+ macrophages were significantly higher in CtD and CtD CT135− infected mice compared to mock infection but did not differ between the two groups of infected mice. We next

investigated if macrophages between groups differed in M1 (MHCII, iNOS, and TNF-α) and M2 (CD206, Arginase, and IL-10) phenotype markers. We found CtD infected mice exhibited significantly higher expression of TNF-α in the population of Ly6C+CD11b+F4/80+ macrophages than CtD CT135− infected mice at day 56 pi (Fig. 1g, h). TNF-α and MHC II expression were

**Fig. 2 Innate immunity is sufficient to drive CT135 dependent immunopathology. a** Infection of RAG1$^{-/-}$ mice with CtD and CtD CT135$^-$, one mouse per panel is depicted. **b** Gross genital tract pathology of RAG1$^{-/-}$ infected mice at day 56 pi. **c** Quantification of gross uterine horn pathology of RAG1$^{-/-}$ infected mice at day 56 pi. ($n = 7$ mice per group). **d** Histopathology and immunohistochemistry of RAG1$^{-/-}$ infected mice at day 56 pi. Images shown are representative of $n = 3$ mice per condition. **e** Flow cytometry of total CD11b$^+$ F4/80$^+$ macrophages in the whole GT of CtD and CtD CT135$^-$ infected mice at day 56 pi ($n = 5$ and 4 for CtD and CtD CT135$^-$ infection respectively). **f** Flow cytometry of total CD11b$^+$ F4/80$^+$ Ly6C$^+$ TNF-α$^+$ macrophages in the whole GT of CtD and CtD CT135$^-$ infected mice at day 56 pi. **g** Flow cytometry CD11b$^+$ F4/80$^+$ Ly6C$^+$ TNF-α$^+$ macrophages in panel (**g**) ($n = 5$ and 4 for CtD and CtD CT135$^-$ infection respectively, one-tailed student's t test). **h** Infection kinetics of CtD infected C57BL/6 and CCR2$^{-/-}$ mice. ($n = 10$ mice per group). **i** Infection kinetics of CtD CT135$^-$ infected C57BL/6 and CCR2$^{-/-}$ mice. ($n = 10$ mice per group). **j** Quantification of gross uterine horn pathology at day 56 pi. ($n = 10$ mice per group). **c**, **e**, **h**–**j** Two-tailed Mann–Whitney test. **c**, **e**, **f**, **h**–**j** Data are pooled from two independent experiments and shown as the mean ± SD.

---

the major differences between macrophages from CtD and CtD CT135$^-$ infected mice (Supplementary Figs. 3 and 4). We next performed chlamydial 16s mRNA in situ hybridization (ISH) staining of mock, CtD and CtD CT135$^-$ infected GT tissue. Chlamydial infection was restricted to uterine epithelial cells and luminal neutrophils. Notably, chlamydial 16 s mRNA was not detected in submucosal tissues (Fig. 1i) despite the intense infiltration of IBA1$^+$ macrophages.

**Innate immunity is sufficient to cause CT135 dependent GT immunopathology.** Two experimental findings in Fig. 1 provided insight about possible CT135 function: (i) CtD infections produced significantly high burdens early pi implying an important pathogenic function of CT135 in the avoidance of early innate immune factors, and (ii) CtD infection, but not CtD CT135$^{-/-}$ caused macrophage-associated pathology implicating a role for CT135 in immunopathogenesis. To investigate the role of CT135 in the avoidance of innate immune factors, we infected RAG1$^{-/-}$ mice with CtD and CtD CT135$^-$ and monitored infection and pathology. CtD and CtD CT135$^{-/-}$ infection of female RAG1$^{-/-}$ mice resulted in persistent infections; however, the cervicovaginal shedding profiles were distinctly different (Fig. 2a). CtD infection resulted in continuous cervicovaginal positive cultures with varying degrees of infectious burden throughout the entire culture period. In contrast, CtD CT135$^-$ infection produced a very different shedding profile characterized by intermittent episodes of culture positivity and negativity. Importantly, CtD CT135$^-$ infected mice exhibited a marked reduction in severe GT pathology at day 56 pi. GT pathology in CtD infected RAG1$^{-/-}$ animals was comparable to that observed in CtD infected C57BL/6 mice shown by the severe gross uterine horn and uterine glandular dilation associated with the infiltration of endometrial submucosal macrophages (Fig. 2b–d). The total number of macrophages from CtD and CtD CT135$^-$ infected mouse GT did not differ between groups (Fig. 2e). Importantly, Ly6C$^+$CD11b$^+$ F4/80$^+$ macrophages isolated from CtD infected GT expressed significantly higher levels of TNF-α (Fig. 2f, g). The expression level of MHCII, iNOS, CD206, and IL-10 did not differ between macrophages from the two groups (Supplementary Fig. 5). These results demonstrate that innate immunity is sufficient to cause CT135 mediated pathology, but pathology required continuous shedding from the GT. The histopathological findings implicated a role for macrophages in the immunopathogenesis of endometrial disease. Monocyte emigration from bone marrow to external tissues during bacterial infection is mediated by the chemokine receptor CCR2[30]. To determine if CCR2 was required for CT135 mediated pathology, CCR2 KO mice were infected with CtD and CtD CT135$^-$. Despite nearly identical infection kinetics, infected CCR2 KO mice exhibited significantly less uterine horn pathology than C57BL/6 (Fig. 2h–j). We conclude from these findings that CT135 is the key virulence factor for

evading host innate immunity and is required for driving macrophage-associated GT immunopathology.

**CT135 evades innate host defense by triggering neutrophil cytotoxicity.** To identify which cellular component of innate immunity provided the mechanistic link between CT135, infectivity, and the development of damaging immunopathology, we performed CtD CT135$^-$ rescue experiments in gene-deficient or antibody-depleted RAG$^{-/-}$ mice. These experiments were designed to identify treatments capable of specifically rescuing the infectivity of the CtD CT135$^-$ strain. Innate immunity gene KO or antibody-treated RAG1$^{-/-}$ mice were infected transcervical with CtD or CtD CT135$^-$ and cervicovaginal chlamydial loads were determined at day 7 pi. As neutrophils are known to be important in innate immunity control of murine chlamydial GT infection[31], we first depleted neutrophils with Ly6G antibody. We found treatment with anti-Ly6G specifically rescued the infectivity CtD CT135$^-$ (Fig. 3a). In contrast, liposomal clodronate depletion of macrophages had no effect on CtD CT135$^-$ infection (Fig. 3b). Mice deficient in ILC, perforin, or IFN-γ showed a significant increase in chlamydial loads demonstrating the importance of these innate effectors in anti-chlamydial innate immunity; however, these effects were not specific to CtD CT135$^-$ (Fig. 3c–e). Blocking the type I IFN receptor did not affect the infectivity of either strain (Fig. 3f). Collectively, these findings provide evidence that CT135 targets neutrophils to avoid host innate immunity.

To investigate the mechanism by which CT135 evaded neutrophil host defense, we studied CtD and CtD CT135$^-$ neutrophil interactions. Mouse bone marrow-derived neutrophils (BMDN) were infected with CtD and CtD CT135$^-$ neutrophil at different MOI and cytotoxicity monitored by lactate dehydrogenase (LDH) release. Neutrophil cytotoxicity was Ct135 and MOI dependent (Fig. 3g). CT135 cytotoxicity (Fig. 3h) and IL-1β secretion (Fig. 3i) were inhibited in CtD by rifampicin, an inhibitor of de novo chlamydial mRNA synthesis. Both cytotoxicity and IL-1β secretion, two CT135 dependent phenotypes, were complemented when the CtD CT135$^-$:: bla CT135 strain was used to infect BMDN (Fig. 3h, i). TEM of CtD and CtD CT135 infected neutrophils showed both strains confined within multiple phagocytic vacuoles (Fig. 3j). Structurally, chlamydial organisms of both strains were contained in vacuoles and resembled small EB to RB transitional forms or larger RBs, findings in agreement with previous reports of chlamydial neutrophil interactions[32]. Importantly, although similar developmental transition forms were common to both chlamydial strains, only infection with CtD resulted in neutrophil cytotoxicity and IL-1β secretion that was dependent on chlamydial mRNA synthesis. Collectively, the findings suggest CT135 functions as an IM pore protein that secretes PAMP(s) into the neutrophil cytosol resulting in IL-1β secretion and lytic cell death.

**CT135 activates neutrophil NLRP3/Caspase1 inflammasome through the TLR2/MyD88 signaling pathway.** To identify the CT135 mediated IL-1β signaling pathway, we first screened a panel of bone marrow-derived macrophage (BMDM) deficient in multiple signaling pathways. We found CtD CT135 dependent secretion of IL-1β was not detected in TLR2$^{-/-}$, MyD88$^{-/-}$, NLRP3$^{-/-}$, and IL-1β$^{-/-}$IL-18$^{-/-}$ BMDM, demonstrating that

the NLRP3/Caspase1 pathway inflammasome through the TLR2/MyD88 signaling pathway is essential for CT135 mediated IL-1β secretion (Supplementary Fig. 6).

We next corroborate these findings using bone marrow-derived neutrophils (BMDN). IL-6 was not detected in CtD infected TLR2$^{-/-}$ and MyD88$^{-/-}$ BMDN. CtD CT135$^-$ infected BMDN derived from these same KO mice did not produce

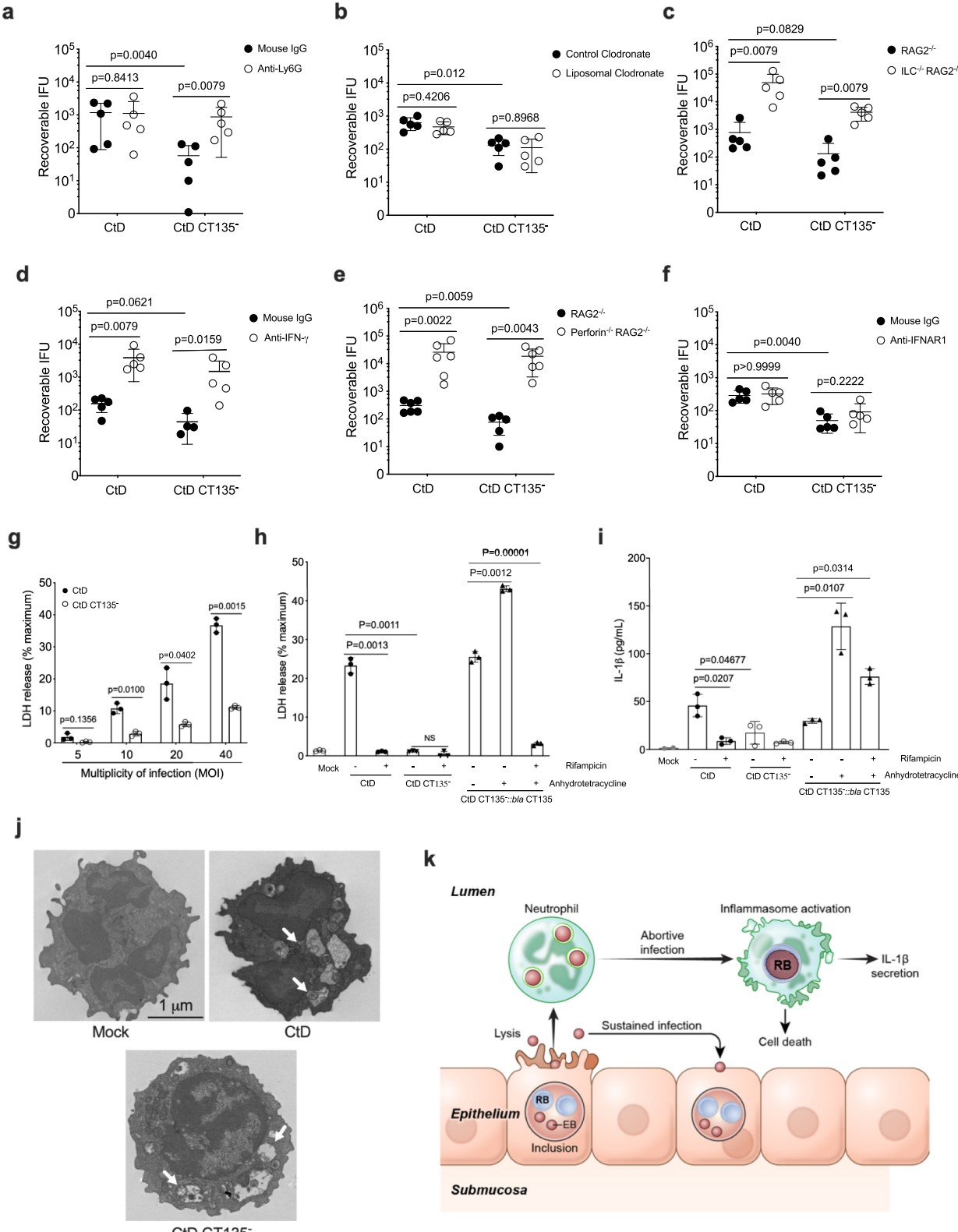

**Fig. 3 CT135 induces neutrophil cytotoxicity to evade innate host defense. a–f** Knock out mice or antibody-treated RAG2$^{-/-}$ mice were infected with CtD or CtD CT135$^{-}$ at day 7 pi. **a** Mouse IgG and anti-Ly6G. **b** Control clodronate and liposomal clodronate. **c** RAG2$^{-/-}$ and ILC$^{-/-}$RAG2$^{-/-}$ mice. **d** Mouse IgG and anti-IFNγ. **e** RAG2$^{-/-}$ and Perforin$^{-/-}$RAG2$^{-/-}$ mice. **f** Mouse IgG and anti-IFN-α/β R1. Depletion of neutrophils specific rescued the infectivity of CtD CT135$^{-}$. (**a–f** $n = 5$ mice per group, two-tailed Mann–Whitney test). **g** LDH release by bone marrow-derived neutrophils (BMDN) infected with different multiplicity of infection (MOI) of CtD and CtD CT135$^{-}$. (Two-tailed unpaired student's $t$ tests). **h** LDH release in CtD infected BMDN is dependent on chlamydial de novo mRNA synthesis. **i** IL-1β secretion of BMDN infected with CtD, CtD CT135$^{-}$ and CtD CT135$^{-}$:: *bla CT135*. Anhydrotetracycline hydrochloride (aTC) (50 ng/ml) was added to BMDN prior- and post-infection for the induction of CT135. (**h, i** two-tailed Mann–Whitney test). **g–i** Result shown are representatives of two independent experiments. **j** Transmission electron micrographs of mock, CtD, and CtD CT135$^{-}$ infected BMDN. Arrows denote inclusions containing chlamydiae undergoing early partial developmental reorganization. Pictures shown are representatives of two independent experiments. **k** Schematic model of chlamydial infection of uterine epithelial cells and interaction with neutrophils as a mechanism to evade neutrophil host defense. Lysed infected epithelial cells release numerous infectious EBs that are phagocytized by luminal neutrophils. Infection of neutrophils is abortive resulting in partial chlamydial development that is sufficient to cause cytotoxic death and promote IL-1β secretion. The ability to kill neutrophils allows CtD organisms to evade neutrophil host defense and sustain chronic infection of the uterine epithelial cells. **a–i** Data are shown as the mean ± SD.

significant amounts of IL-6 (Fig. 4a). IL-1β was not detected in CtD infected TLR2$^{-/-}$, MyD88$^{-/-}$, NLRP3$^{-/-}$, and Casp1$^{-/-}$ BMDN. CtD CT135$^{-}$ infected BMDN derived from these same KO mice did not produce significant amounts of IL-1β (Fig. 4b, c).

To validate these in vitro findings, we infected female C57BL/6, TLR2$^{-/-}$, MyD88$^{-/-}$, NLRP3$^{-/-}$, and Casp1$^{-/-}$ knockout mice with CtD and determined cervicovaginal chlamydial loads and gross uterine pathology. CtD infected TLR2$^{-/-}$, MyD88$^{-/-}$, NLRP3$^{-/-}$, and Casp1$^{-/-}$ KO mice exhibited significantly greater infectious loads at 7 pi, implicating their importance in controlling GT infection (Fig. 4d). Remarkably, infection of knockout mice showed distinct differences in GT pathology (Fig. 4e). CtD infected TLR2$^{-/-}$ and MyD88$^{-/-}$ mice exhibited severe gross uterine horn pathology while NLRP3$^{-/-}$ and Casp1$^{-/-}$ mice had minimal to no uterine horn pathology. Collectively, we conclude from these experiments that CtD CT135 activates the NLRP3/Caspase1 inflammasome through the TLR2/MyD88 signaling pathway, initiating neutrophil cytotoxicity to evade innate immunity (Fig. 4f). In addition, the results show that CT135 activation of the NLRP3/Caspase1 inflammasome is sufficient to drive macrophage-associated GT pathology, a paradoxical finding because chlamydiae were absent in the submucosal macrophages (Fig. 1i).

**GT immunopathology is dependent on NLRP3/Caspase1/IL-1α activation.** To further investigate NLRP3 inflammasome mediated GT immunopathology, NLRP3$^{-/-}$, Casp1/11$^{-/-}$, IL-1α$^{-/-}$, IL-1β$^{-/-}$IL-18$^{-/-}$, and IL-1R$^{-/-}$ mice were infected with CtD. Pathology and infection were monitored. CtD infected NLRP3$^{-/-}$ mice exhibited minimal to no uterine horn pathology at day 56 pi (Fig. 5a) and produced equivalent numbers of pups to mock-infected mice (Fig. 5b). CtD infected Casp1/11$^{-/-}$ and IL-1α$^{-/-}$ mice exhibited minimal to no uterine horn pathology (Fig. 5c, d) at day 56 pi. In contrast, IL-1β$^{-/-}$IL-18$^{-/-}$ and IL-1R$^{-/-}$ mice had severe gross uterine horn disease (Fig. 5e, f) at day 56 pi. The IL-1α findings are paradoxical but are consistent with reports that IL-1α is capable of functioning as an intracrine proinflammatory activator[33].

CtD infected NLRP3$^{-/-}$, Casp1/11$^{-/-}$, IL-1β$^{-/-}$IL-18$^{-/-}$, and IL-1R$^{-/-}$ mice all exhibited significantly greater infectious loads at day 3 and 7 pi showing NLRP3/Caspase1 inflammasome signaling pathway is critical for controlling early chlamydial infection (Fig. 5g–k). IL-1α$^{-/-}$ mice exhibited a significantly higher burden throughout GT infection. These findings show IL-1α plays an important role in controlling *C. trachomatis* infection of uterine epithelial cells that is independent of the IL-1R.

Our findings that CT135 dependent NLRP3 inflammasome activation is critical for pathology development agree with a

previous report[34]. NLRP3 dependent pathology was independent of infection duration but strongly associated with higher early infectious loads. Because GT pathology occurred late post infection, we propose early high infection loads trigger NLRP3-dependent innate immune programming which is important for the development of late pathology. Furthermore, despite low infectious burdens, there was an abundance of chlamydial infected luminal neutrophils detected by ISH at day 56 pi. CtD infection, but not CtD CT135$^{-}$ infection, activates the NLRP3 inflammasome in neutrophils in Fig. 4. Macrophages acquire trained memory-like characteristics[35] and therefore this trained late-stage immunity could cause the immunopathology in our model.

**GT immunopathology is P2X7 receptor-dependent.** NLRP3 inflammasome activation triggers caspase-1 activation through a two-signal mechanism[36]. The NLRP3 inflammasome pathway can be activated by diverse stimuli including microbial PAMPs, endogenous molecules, and extracellular ATP[36], and requires priming by transcriptional regulators such as TNF-α[37]. Localization of chlamydiae in infected GT tissue by 16s mRNA ISH showed chlamydial organisms were limited to the uterine epithelium and luminal neutrophils (Fig. 1h). This was a paradoxical finding as it was inconsistent with an infection-dependent stimulus that was driving the activation of the NLRP3 inflammasome in infiltrating macrophages.

To determine whether chlamydial PAMPs or endogenous guanylate binding protein (GBP)[38] activated inflammasome mediated endometrial disease, NLRC4$^{-/-}$ and AIM2$^{-/-}$, NOD1$^{-/-}$, NOD2$^{-/-}$, and GBP$^{chr3-/-}$ mice were infected with CtD. We found NLRC4$^{-/-}$ and AIM2$^{-/-}$, NOD1$^{-/-}$, NOD2$^{-/-}$, and GBP$^{chr3-/-}$ mice developed severe endometritis at day 56 pi (Supplementary Figs. 7a–e). As shown in Fig. 4e, MyD88$^{-/-}$ mice also developed pathology. Collectively these results suggest chlamydial PAMPs and GBP are not activating NLRP3 inflammasome and causing endometrial disease.

Therefore, we hypothesized extracellular ATP generated from infected epithelial cells[39] and neutrophils might serve as a DAMP to activate the NLRP3 inflammasome in submucosal macrophages. We infected primary murine oviduct epithelial cells with CtD and assayed extracellular ATP concentrations in culture supernatants at different times pi. Micromolar concentrations of ATP were released from CtD infected oviduct epithelial cells in increasing amounts over the 48 h culture period (Fig. 6a). We further quantitated ATP release from BMDN infected with CtD and CtD CT135$^{-}$. ATP released from CtD infected BMDN cells was significantly increased compared to CtD CT135$^{-}$ infected BMDN at 2 and 4 hpi (Fig. 6b). Extracellular ATP concentrations from CtD infected BMDN ($2 \times 10^5$ cells) was 2 μM and peaked at

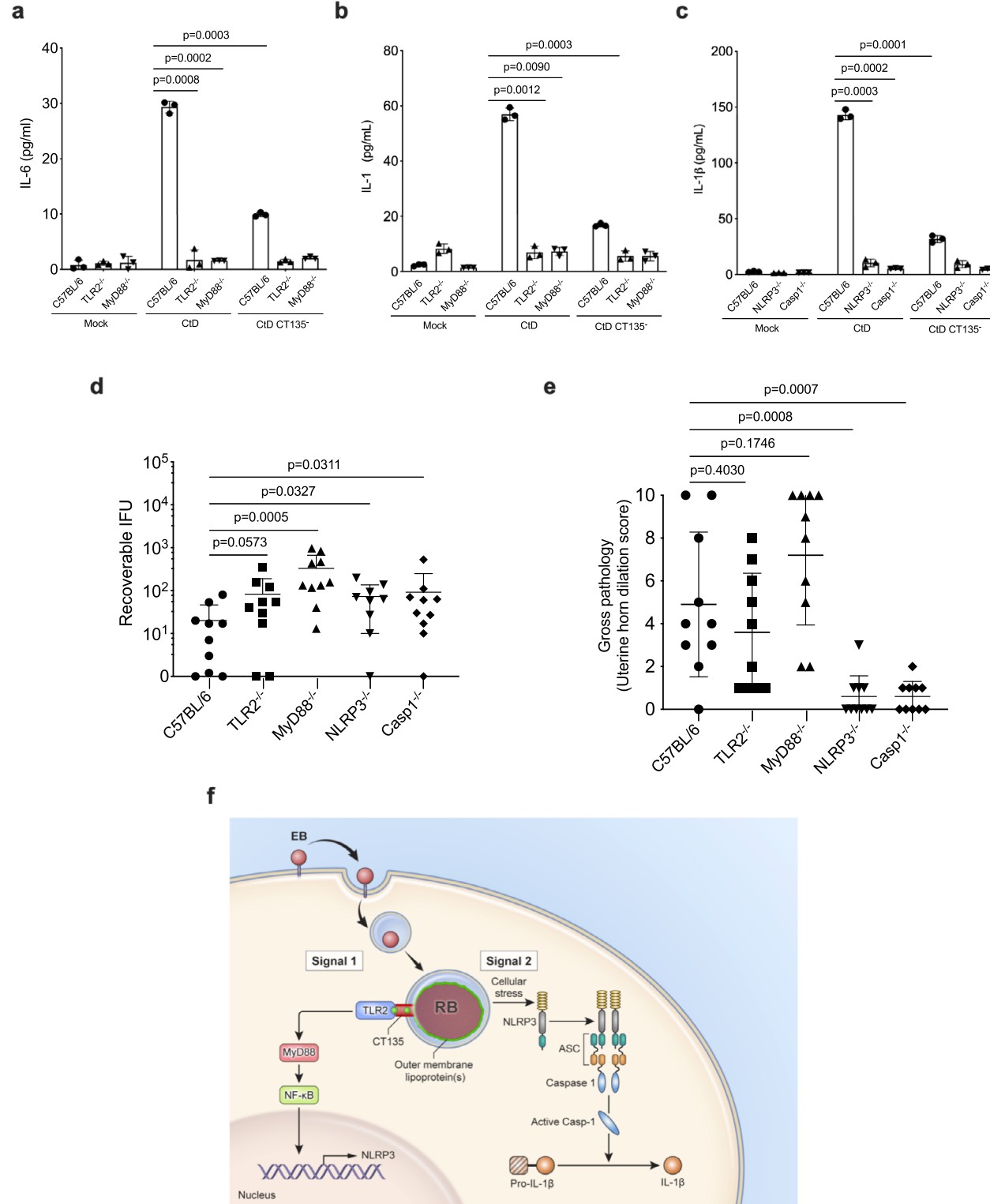

2 h pi. Extracellular ATP concentrations were significantly reduced in CTD CT135$^-$ infected BMDN.

We next tested if extracellular ATP produced from CtD infected BMDN activated NLRP3 inflammasome in BMDM. BMDN were infected with CtD or CtD CT135$^-$, supernatants were collected and then incubated with TNF-α primed BMDM. NLRP3 inflammasome activation in primed BMDM was assayed for IL-

1β by ELISA. IL-1β secretion in BMDM was detected from the supernatants of CtD but not CtD CT135$^-$ infected BMDN. CtD CT135 dependent IL-1β secretion in NLRP3$^{-/-}$, Caspase1$^{-/-}$, and P2X7R$^{-/-}$ BMDM was significantly reduced. The CtD CT135$^-$ phenotype was restored following complementation with the CtD CT135$^-$::*bla CT135* infected BMDN as shown by a significant increase of IL-1β secretion (Fig. 6c). These results show

**Fig. 4 CT135 activates the NLRP3 inflammasome through the TLR2/MyD88 signaling pathway. a** Secretion of IL-6 in CtD and CtD CT135⁻ infected BMDN derived from C57BL/6, TLR2⁻/⁻, MyD88⁻/⁻ mice ($n = 3$ mice per group). **b** Secretion of IL-1β in CtD and CtD CT135⁻infected BMDN derived from C57BL/6, TLR2⁻/⁻, MyD88⁻/⁻ mice ($n = 3$ mice per group). **c** Secretion of IL-1β in CtD and CtD CT135⁻ infected BMDN derived from C57BL/6, NLRP3⁻/⁻, Caspase1⁻/⁻ mice ($n = 3$ mice per group). (**a–c** two-tailed unpaired student's $t$ tests). **a–c** Result shown are representatives of two independent experiments. **d** CtD infections of TLR2⁻/⁻, MyD88⁻/⁻, NLRP3⁻/⁻, Caspase1⁻/⁻ mice were significantly higher at day 7 pi. ($n = 10$ mice, pooled from two independent experiments, two-tailed Mann–Whitney test). **e** Quantification of gross uterine horn pathology of CtD infected C57BL/6, TLR2⁻/⁻, MyD88⁻/⁻, NLRP3 ⁻/⁻, Caspase1⁻/⁻ infected mice at day 56 pi ($n = 10$ mice per group, pooled from two independent experiments, two-tailed Mann–Whitney test). **f** A model of CT135 dependent neutrophil killing through NLRP3 inflammasome via TLR2/MyD88 signaling. Signal one is triggered by an abortive CtD infection in phagocytes resulting EB to RB transition with an accompanying degradation of chlamydial OM lipoprotein(s)[42]. We hypothesize CT135 exports degraded OM lipoprotein engaging inclusion membrane TLR2[43]. This process results in signaling by MyD88 and NF-κB to upregulate NLRP3 and pro-IL-1β expression. Signal 2 activation of NLRP3 is the result of infection-induced cellular stress[34]. This CtD phagocyte interaction activates the NLRP3 inflammasome resulting in phagocytic cell death and IL-1β secretion. **a–e** Data are shown as the mean ± SD.

extracellular ATP produced from CtD infected BMDN activated NLRP3 inflammasome in BMDM through the P2X7R.

TNF-α mediates sensitization to ATP via the NLRP3 inflammasome in the absence of microbial stimulation[37]. To determine whether the P2X7R signaling pathway contributes to the CtD GT pathology, P2X7R⁻/⁻ and TNFR1/2⁻/⁻ mice were infected and assessed for infection and pathology.

P2X7R⁻/⁻ infected mice had significantly greater infection loads at days 3, 7pi compared to infected C57BL/6 mice, but loads were similar at later times pi (Fig. 6d). Infection burdens in TNFR1/2⁻/⁻ mice were not significantly different compared to C57BL/6 mice throughout the culture period (Fig. 6e). Thus, TNF-α has a minimal role in controlling chlamydial infection of uterine epithelial cells as previously shown[40]. Notably, despite supporting near-normal chlamydial GT infectious loads, P2X7R⁻/⁻ and TNFR1/2⁻/⁻ mice exhibited significantly reduced GT pathology (Fig. 6f), showing the P2X7R signaling pathway is critical for the development of CtD GT pathology.

To determine whether macrophage recruitment was similar among experimental groups, the total number of macrophages from GT tissue of CtD infected mice was analyzed by flow cytometry. Total CD11b⁺ F4/80⁺ macrophages were significantly higher in CtD infected C57BL/6, TNFR1/2⁻/⁻, and P2X7R⁻/⁻ mice compared to mock infection but did not differ among the infected mice (Fig. 6g), excluding the possibility that GT pathology observed in TNFR1/2⁻/⁻ and P2X7R⁻/⁻ mice (Fig. 6f) were not due to differences in macrophages recruitment.

We next phenotyped Ly6C⁺CD11b⁺F4/80⁺ macrophages isolated from CtD infected mice by flow cytometry. C57BL/6, TNFR1/2⁻/⁻ and P2X7R1/2⁻/⁻ mice GT expressed significantly higher levels of TNF-α (Fig. 6h, i). The expression level of MHCII, iNOS, CD206, and IL-10 did not differ between macrophages from the groups (Supplementary Fig. 8). Despite the production of TNF-α in Ly6C⁺CD11b⁺ F4/80⁺ macrophages in TNFR1/2⁻/⁻ and P2X7R⁻/⁻ mice (Fig. 6h, i), infected mice failed to develop uterine horn pathology. These findings are consistent with a two-signal model for NLRP3 inflammasome activation[36].

Overall, the results support the conclusion that chlamydial GT immunopathology is mediated by NLRP3 inflammasome via the P2X7R signaling pathway (Fig. 7). We propose ATP from infected neutrophils and epithelial cells serve as a DAMP driving NLRP3 inflammasome activation. Thus, this represents a model of sterile inflammation in the immunopathogenesis of chlamydial diseases.

## Discussion

Chronic chlamydial infection that drives damaging inflammatory disease is a fundamental pathophysiologic feature of blinding trachoma and PID, but the underlying virulence factors and

pathogenic mechanisms are poorly understood. Here, we report using a female mouse GT infection model to identify both a virulence factor and a pathogenic mechanism that results in chronic infection and damaging inflammatory disease. We found CT135 activates the NLRP3 inflammasome through TLR2/ MyD88 signaling pathway as a pathogenic strategy to evade neutrophil host defense, however, this strategy conjointly caused NLRP3 dependent macrophage-associated endometritis. Paradoxically, NLRP3 inflammasome activation in macrophages occurred independently of macrophage infection.

Chlamydial infection was restricted to epithelial cells and luminal neutrophils. We found that chlamydial infected epithelial cells and neutrophils released extracellular ATP, a DAMP in the activation of the NLRP3 inflammasome. Importantly, P2X7R deficient mice failed to develop CT135 dependent macrophage-associated disease. This finding supports the conclusion that chlamydial infected neutrophils and uterine epithelial cells are a source of extracellular ATP that acts as a DAMP for activating the inflammasome of submucosal macrophages in the absence of chlamydial infection. Although the P2X7R has been shown to play a role in the sterile inflammation of numerous non-infectious chronic inflammatory diseases[22], to our knowledge this is a paradigm of sterile inflammation in the immunopathogenesis of an infectious disease.

Inflammasome activation in neutrophils was CT135 dependent and required de novo chlamydial mRNA synthesis. Ultra-structurally, infected neutrophils contained chlamydial-laden vacuoles with developmental structures like early EB-RB transitional forms. As neutrophils do not support productive chlamydial growth, we believe these organisms represent abortive infections that retained the ability to activate the inflammasome. As CT135 mediated NLRP3 inflammasome activation required TLR2/MyD88 signaling, chlamydial lipoprotein(s) were a logical PAMP candidate for priming the inflammasome. The chlamydial outer membrane possesses several lipoprotein species; the most relevant of these to this work is the small cysteine-rich protein (12 kDa) omcA. OmcA possesses a predicted signal peptidase II-processing site and shares a similar structure to the typical Braun lipoprotein of *Escherichia coli*, known to be a potent TLR2 agonist[41]. Notably, during normal chlamydial development omcA is degraded from the outer membrane early as 2 h pi, a time when EBs are undergoing a transition to RBs[42]. Taken together, we conclude that CT135 is a pore protein that transports degraded omcA lipoprotein fragments from the inclusion to the host cytosol. As TLR2 is known to localize to the IM[43], it would be positioned to bind exported omcA fragments capable of activating the MyD88 transduction signaling pathway. Interestingly, CT135 dependent transport of lipoprotein likely occurs in epithelial cells as part of normal EB to RB transitioning; however, unlike phagocytic cells, NLRP3 is not expressed in endometrial tissue[44], thereby the NLRP3 activating function of CT135 would

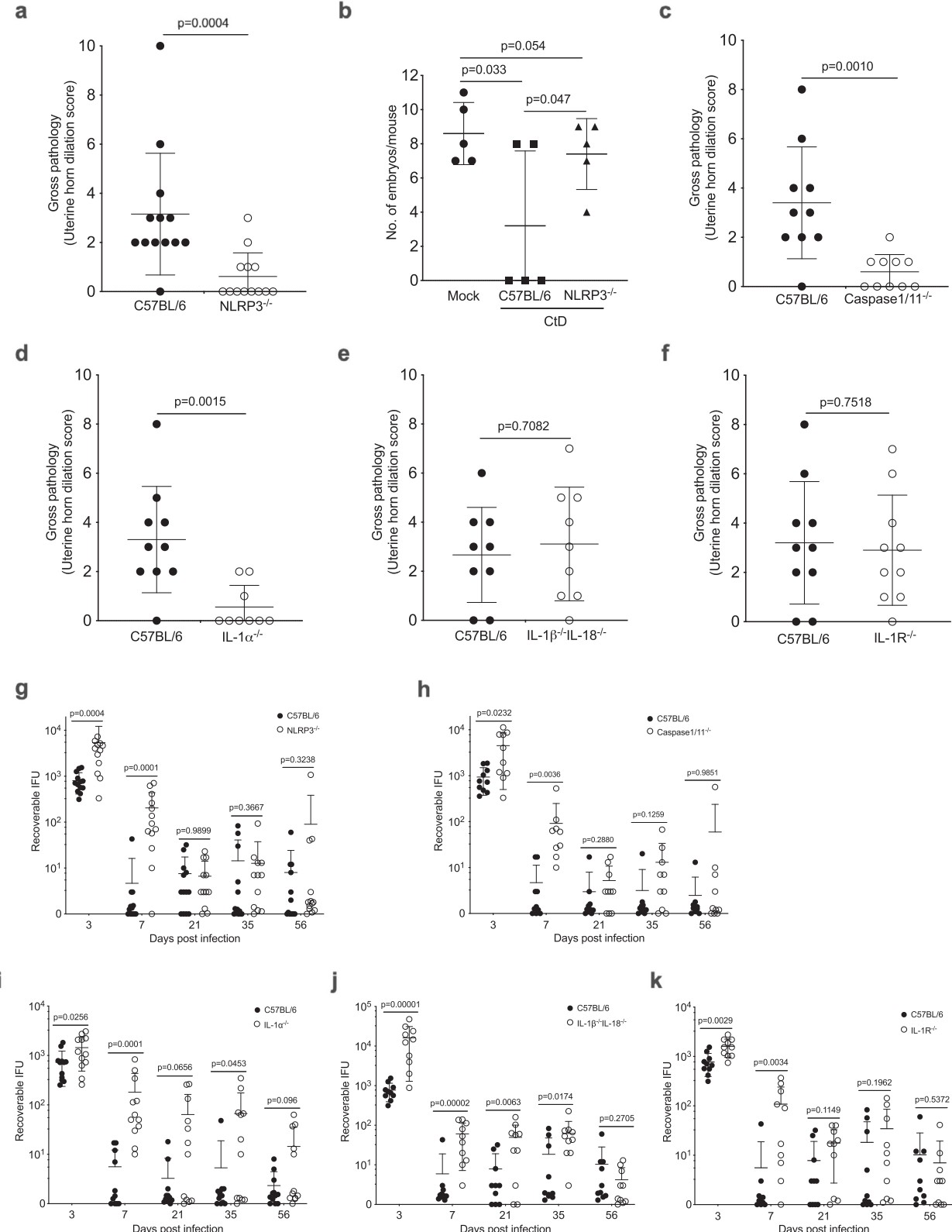

be abrogated in *Chlamydia*'s natural host. Thus, *Chlamydia*'s ability to kill neutrophils as a pathogenic strategy to evade innate immunity evolved by taking advantage of their unique developmental biology.

Cytokine and myeloid inflammatory genes are biomarkers linked to the severity of trachoma and PID leading investigators to propose that innate immunity plays a dominant role in the immunopathogenesis of these diseases[8]. Our finding that innate immunity alone is sufficient to cause macrophage-associated chronic endometritis and infertility in the mouse GT provides compelling evidence in support of innate immunity as a dominant factor in the immunopathogenesis of chlamydial

**Fig. 5 Genital tract immunopathology is dependent on NLRP3/Caspase 1/IL-1α activation. a** Quantification of uterine horn gross pathology in CtD infected C57BL/6 and NLRP3$^{-/-}$ mice at day 56 pi ($n = 13$ mice per group). **b** Mouse Fertility at week 6 pi ($n = 5$ mice per group). **c** Quantification of uterine horn gross pathology in C57BL/6 and Caspase1/11$^{-/-}$ mice at day 56 pi ($n = 10$ mice per group). **d** Quantification of uterine horn gross pathology in C57BL/6 and IL-1α$^{-/-}$ mice at day 56 pi ($n = 9$ mice per group). **e** Quantification of uterine horn gross pathology in C57BL/6 and IL1β$^{-/-}$ IL18$^{-/-}$ mice at day 56 pi ($n = 9$ mice per group). **f** Quantification of uterine horn gross pathology in C57BL/6 and IL-1R$^{-/-}$ mice at day 56 pi ($n = 10$ mice per group). Note that CtD infected NLRP3$^{-/-}$, Caspase 1$^{-/-}$, and IL-1α$^{-/-}$ mice showed significantly reduced oviduct pathology compared to infected C57BL/6 mice. Pathology was abrogated in IL-1α$^{-/-}$ mice but independent of IL-1 receptor. (**a–f** two-tailed Mann–Whitney test). **g–k** Infection kinetics. **g** C57BL/6 and NLRP3$^{-/-}$ mice ($n = 13$ mice per group). **h** C57BL/6 and Caspase1/11$^{-/-}$ mice ($n = 10$ mice per group). **i** C57BL/6 and IL-1α$^{-/-}$ mice ($n = 9$ mice per group). **j** C57BL/6 and IL1β $^{-/-}$ IL18$^{-/-}$ mice ($n = 9$ mice per group). **k** C57BL/6 and IL-1R$^{-/-}$ mice ($n = 10$ mice per group). (**g–k**, two-tailed Mann–Whitney test). **a–k** Data are shown as the mean ± SD. **a, c–k** Data are pooled from two independent experiments.

inflammatory diseases. Macrophage-associated chlamydial GT immunopathology was dependent on NLRP3/Caspase1/IL-1α activation, but interestingly independent of the IL-1R. These findings are consistent with reports of genetic polymorphisms in NLRP3 inflammasome genes in chlamydial PID patients[16]. Polymorphisms in the NLRP3 allele, but not IL-1β or IL-1R alleles, were associated with more severe post-infection sequelae and increased risk of PID tubal pathology[45]. Notably, IL-1α polymorphisms are associated with many chronic inflammatory diseases[46]. It would be interesting to investigate the effect of IL-α polymorphisms in human chlamydial diseases. We also found NLRP3 macrophage inflammasome activation was regulated by TNF-α. Likewise, polymorphisms in the TNF locus are linked with an increased risk of severe inflammatory sequelae in blinding trachoma and PID[13–15]. Collectively, our findings using the mouse GT infection model are in good agreement with genetic risk factors for human chlamydial disease severity, supporting the relevance of the mouse model for the study of human infection and disease. Chronic macrophage-associated endometritis in the mouse did not progress to fibrosis, a pathologic characteristic of both PID and trachoma. The progression of fibrosis requires pro-fibrotic phenotype factors. We observed an M1 pro-inflammatory phenotype at day 56 pi however this time period might not be sufficient to progress to pro-fibrotic macrophages. For example, fibrotic scarring in trachoma is an end-stage disease requiring many years to develop[5].

NLRP3 inflammasome activation triggers caspase-1 activation through a two-signal mechanism. Signal one primes NLRP3 and pro-IL-1β expression by microbial components, IL-1 or TNF-α signaling pathways. Signal two is triggered by microbial products, endogenous molecules, and extracellular ATP to promote NLRP3 inflammasome assembly and caspase-1 activation[36]. Only minimal chlamydial 16s mRNA signals were detected by ISH in submucosal macrophages of chronically infected mice, indicating NLRP3 macrophage inflammasome activation occurred independently of chlamydial infection. To exclude a role for chlamydiae in NLRP3 macrophage inflammasome activation we tested a panel of inflammasome-associated gene-deficient mice for endometrial disease. As previously reported, we found TLR2$^{-/-}$[47], MyD88$^{-/-}$[48], NOD1$^{-/-}$[49], NOD2$^{-/-}$, GBP$^{-/-}$ knockout mice developed severe endometritis, excluding them in the activation of NLRP3 inflammasome. In contrast, it has been reported TLR2 is important in the development of oviduct pathology[50]. The reason for this discrepancy in findings is not clear. We, therefore, reasoned that a DAMP, such as extracellular ATP, generated from neutrophils and epithelial cells (Fig. 6a, b) activated the NLRP3 inflammasome through the P2X7R signaling pathway. P2X7R deficient infected mice developed minimal to no GT pathology demonstrating a role for ATP in activating the NLRP3 inflammasome in sub-mucosal macrophages. Others have previously tested P2X7R deficient mice using the mouse model

but found it had no effect on GT pathology[51], a finding inconsistent with our results. A reason for this discrepancy could be those investigators used P2X7R deficient mice generated on a DBA/2 genetic background, a strain that is known to be resistant to chlamydial GT pathology[52]. Thus, our findings support a mechanism of sterile inflammation in the immunopathology of chlamydial infection such as PID and blinding trachoma. Although NLRP3 sterile inflammation is known to function in several chronic diseases[22], to our knowledge our findings describe a model of sterile inflammation in the immunopathogenesis of an infectious disease.

## Methods

**Chlamydiae**. A low passage plaque cloned *C. trachomatis* D/UW-3/Cx strain (CtD) was isolated[28]. In vitro passage of low passage wild-type, clinical isolates rapidly lead to gene disrupting mutations at the CT135 locus[21]. The CtD CT135$^-$ strain was made by serially passing CtD in McCoy cells (ATCC, catalog number CRL-1696). At the 12th passage, organisms were plaque cloned, grown, and whole-genome sequenced. The CtD CT135$^-$ strain is isogenic to the wild type CtD strain except for a single base transition (C > T) at nucleotide position 152743 leading to Arg201 (CGA) change to a stop codon (TGA). This mutation centrally disrupts the CT135 open reading frame (ORF). The phenotype of the CtD CT135$^-$ strain was confirmed by proteomic analysis (Supplementary Fig. 2). The data have been deposited with links to BioProject accession number PRJNA742023 in the NCBI BioProject database (https://www.ncbi.nlm.nih.gov/bioproject/). To complement the CtD CT135$^-$ phenotype, we made CtD CT135$^-$:: *bla CT135* strain. Briefly, CT135 was PCR amplified from CtD genomic DNA and was cloned into the pBRDUW3 vector[28] containing the tetracycline-inducible promoter. The CT135 ORF was expressed as a Flag tag C-terminal fusion. The integrity of the construct was verified by DNA sequencing. The pBRDUW3-Tet-CT135 vector was then transformed into the plasmid-free CtD CT135$^-$ strain. An ampicillin-resistant clone was plaque purified and expanded yielding CtD CT135$^-$:: *bla CT135* which was used for complementation experiments (Supplementary Fig. 2). Chlamydiae were propagated in McCoy cells and purified by density gradient centrifugation.

**Whole-genome sequence**. DNA was prepared for sequencing using 1 µg of input following the TruSeq DNA PCR-free protocol (Illumina, San Diego, CA). Libraries were sequenced on the Illumina MiSeq instrument as paired-end 2 × 151 base pair reads. Raw fastq reads were trimmed of adapter sequences using cutadapt version 1.12[53] and then trimmed and filtered for quality using the FASTX-Toolkit 0.0.14 (Hannon Lab, CSHL). The remaining reads were mapped to the *Chlamydia trachomatis* genome (NC_000117.1, NC_017435.1) using Bowtie2 version 2.2.9[54] with parameters –local –no-mixed -X 1000. PCR duplicates were removed using picard MarkDuplicates (Broad Institute) and variants were called using GATK HaplotypeCaller version 4.1.2.0[55] with parameter -ploidy 1.

**Mice**. Six to eight-week-old female C57BL/6, Rag1$^{-/-}$ mice (stock number: 146), Rag2$^{-/-}$ mice (stock number: RAGN12-F), Rag2/Il2rg$^{-/-}$ (stock number: 4111-F) deficient in T, B and innate lymphoid cells (ILCs), Pfp/Rag2$^{-/-}$ (stock number: 1177) and TLR3$^{-/-}$ (stock number: 302) were obtained from Taconic Laboratories through an NIAID contract. NLRP3$^{-/-}$ (stock number: 021302), TNF p55$^{-/-}$/p75$^{-/-}$ (TNFR1/2$^{-/-}$; stock number: 003243), P2X7R$^{-/-}$ (stock number: 005576), AIM2$^{-/-}$ (stock number: 013144), Caspase1/11$^{-/-}$ (stock number: 016621), TLR2$^{-/-}$ (stock number: 004650), TLR4$^{-/-}$ (stock number: 007227), TLR7$^{-/-}$ (stock number: 008380), TLR9$^{-/-}$ (stock number: 34329), MyD88$^{-/-}$ (stock number: 009088), NOD2$^{-/-}$ (stock number: 005763), IL-1R$^{-/-}$ (stock number: 003245), TRIF$^{-/-}$ (stock number: 005037), STING$^{-/-}$ (stock number:

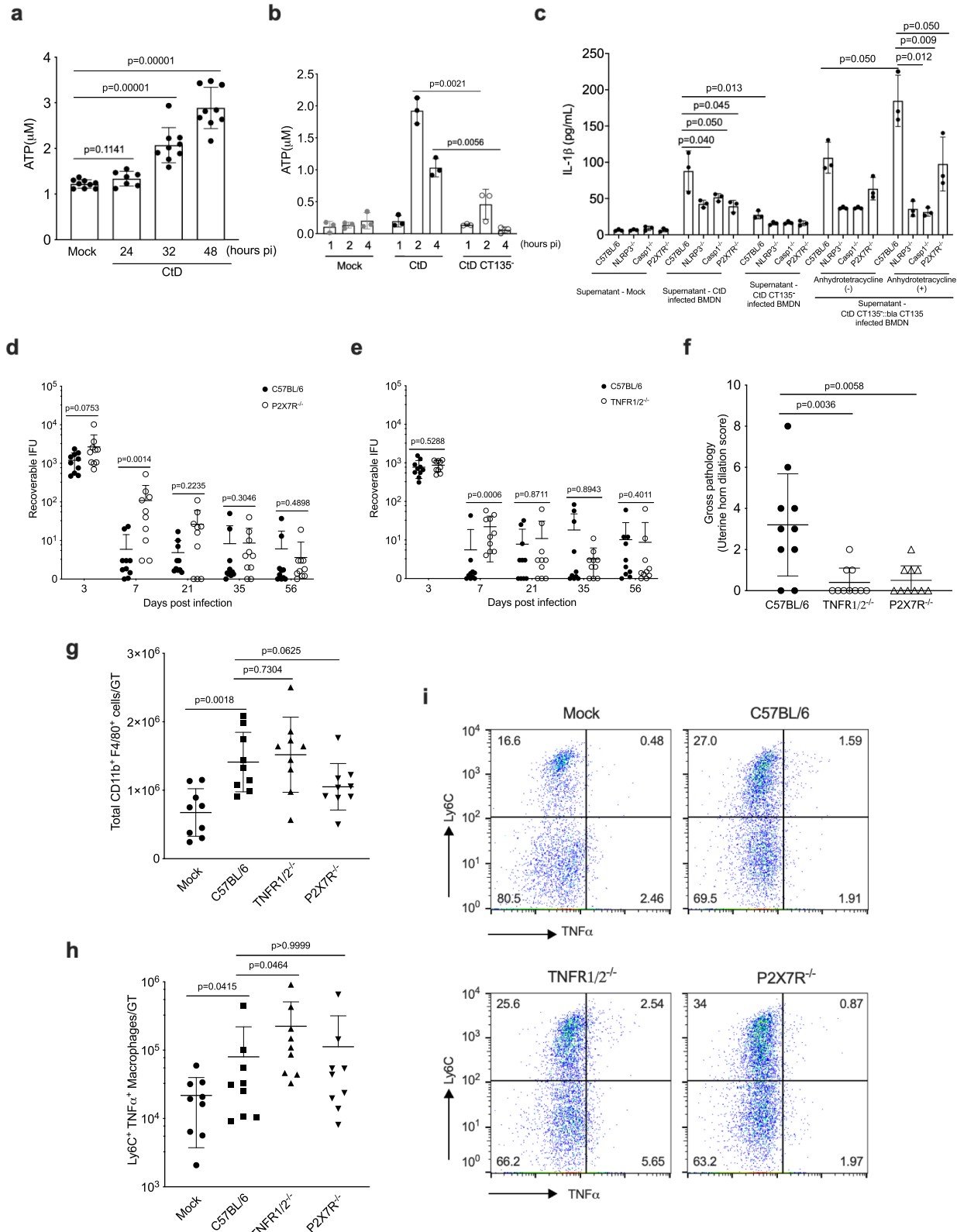

025805), cGAS⁻/⁻ (stock number: 026554) mice were purchased from Jackson Laboratories. NLRC4⁻/⁻ mice were provided by Dr. Gabriel Nunez (University of Michigan). Casp1⁻/⁻ mice were provided by Dr. Thirumala-Devi Kanneganti (St. Jude Children's Research Hospital). IL-1α⁻/⁻ mice were made by Dr. Yoichiro Iwakura (University of Tokyo) and provided by Sunny Shin (University of Pennsylvania). IL-1β⁻/⁻ IL-18⁻/⁻ mice were made by Dr. David Chaplin (University of Alabama) and provided by Dr. Edward Miao (University of North Carolina). GBP^chr3−/− mice were made by Drs. Masahiro Yamamoto and Kiyoshi Takeda and provided by Dr. Jorn Coers (University of Duke). NOD1⁻/⁻ mice were provided by Dr. Dragana Jankovic (NIAID/NIH).

**Murine model of GT infection**. Six to eight-week-old female mice were injected with 2.5 mg of medroxyprogesterone (Depo-Provera; Pharmacia &Upjohn, NY)

**Fig. 6 Genital tract immunopathology is dependent on the P2X7 signaling pathway. a** ATP release from chlamydial infected BM12.4 cells. ATP released from infected cells was significantly increased throughout the course of infection ($n = 9$ per group). **b** ATP release from chlamydial infected BMDN. ATP released from CtD infected BMDN was significantly increased compared to CtD CT135$^-$ infected BMDN cells at 2- and 4-h pi. **c** Secretion of IL-1β in BMDM derived from C57BL/6, NLRP3$^{-/-}$, Caspase1$^{-/-}$, and P2X7R$^{-/-}$ mice, BMDM was primed with TNF-α (100 pg/ml) for 6 h and then treated with the supernatants from CtD, CtD CT135$^-$, and CtD CT135$^-$:: *bla CT135* infected BMDN. Anhydrotetracycline hydrochloride (aTC) (50 ng/ml) was added to BMDN prior to and post-infection for the induction of CT135 ($n = 3$ per group). **b, c** Two independent experiments were performed. **a–c**, one-tailed student's *t* test). **d** Infection of CtD infected C57BL/6 and P2X7R$^{-/-}$ mice ($n = 10$ mice per group). **e** Infection kinetics of C57BL/6 and TNFα p55$^{-/-}$p75$^{-/-}$ mice ($n = 10$ mice per group). **f** Quantification of gross uterine horn pathology at day 56 pi ($n = 10$ mice per group). **g** Flow cytometry of total CD11b$^+$ F4/80$^+$ macrophages in the whole GT of CtD infected mice at day 56 pi ($n = 9$ mice per group). **h** Flow cytometry of total CD11b$^+$ F4/80$^+$ Ly6C$^+$ TNF-α$^+$ macrophages in the whole GT of CtD infected mice ($n = 10$ mice per group). **i** Flow cytometry CD11b$^+$ F4/80$^+$ Ly6C$^+$ TNF-α$^+$ macrophages in panel (**h**). (**d–h** Two-tailed Mann–Whitney test). **a–h** Data are shown as the mean ± SD. **a** Data are pooled from three independent experiments. **d–h** Data are pooled from two independent experiments.

---

subcutaneously at 10 and 3 days prior to chlamydial infection and then transcervical infected with $1 \times 10^5$ inclusion forming units (IFU) of chlamydiae in SPG buffer (10 mM phosphate, pH7.2, containing 0.25 M sucrose and 5 mM L-glutamic acid) by using NSET device. The course of infection is monitored by swabbing the vaginal vault with calcium alginate swabs (Puritan Medical, Guilford, Maine) at selected intervals followed by an enumeration of recoverable IFU on McCoy cell monolayers. Three mice from each group were euthanized at 56 days post infection (pi) and the GTs were removed for Hematoxylin and Eosin, immunohistochemical, and ISH. Mice were randomly assigned to experimental groups. Mice maintained on 12 h light/12 h dark cycles under 18-23 °C with 40-60% humidity in the National Institute of Allergy and Infectious Diseases (NIAID) animal facility. Experiments were performed in agreement with guidelines established by the Institutional Animal Care and Use Committee (ACUC) guidelines for the care and use of laboratory animals and approved by the NIAID ACUC.

**Fertility study.** Mice infected with CtD or CtD CT135$^-$ were caged with male C57BL/6 or NLRP3$^{-/-}$ mice at 6 weeks pi. Pregnancy was monitored daily by a board-certified veterinarian. Once obvious signs of pregnancy were observed, mice were euthanized and the numbers of embryos in both uterine horns were counted. If mice were not pregnant in 8 weeks, mice were cohoused with a different functional male mouse that had fathered a litter during the first mating.

**In vivo mAb treatment.** For in vivo neutrophil depletion, 300 μg anti-L6G clone 1A8 (BioXcell) mAb or control IgG clone 2A3 (BioXcell) were administered intraperitoneally at days −2, 0 and +2, +4, +6 post transcervical challenge. For neutralization of signaling through the IFN-α/β receptor, mice were treated intraperitoneally with 500 μg per mouse of anti-IFNAR1 clone MAR1-5A3 or control IgG clone MOPC-21 (BioXcell) in phosphate-buffered saline (PBS) at days −1 and +1, +3, +5 post transcervical challenge. For neutralization of IFN-γ, 500 μg anti-IFN-γ clone XMG1.2 mAb or control IgG clone TNP6A7 (BioXcell) were administered intraperitoneally at days −1 and +2, +5 post transcervical challenge.

**Depletion of macrophages by Clodronate liposome.** RAG2$^{-/-}$ mice were inoculated with Clodronate liposome or PBS liposome via the intra-orbital sinus (250 μl) and transcervical (5 μl) routes at the days −4, −1 and +2, +5 post intracervical challenge.

**Histopathology, immunohistochemical, and ISH.** Mouse female GTs were collected at 56 days post-infection, fixed in formalin, and processed into paraffin wax blocks. Sections were processed for Hematoxylin and Eosin (H&E) and immunohistochemical (IHC) staining. ISH and IHC staining was carried out on the Bond RX (Leica Biosystems) platform using established vendor protocols. Briefly, 5μm-thick sections were deparaffinized and rehydrated. Heat-induced epitope retrieval was performed using Epitope Retrieval Solution 2 (Leica Biosystems), pH 9.0, at 95 °C for 15 min, followed by an enzyme protease treatment at 40 °C for 15 min. Probe hybridization was performed using the RuneScape 2.5 LS Probe Ctr-16SrRNA (Advanced Cell Diagnostics) for two hours followed by signal amplification steps using AMP1-6 solutions. Detection with Fast Red chromogen was completed using the Bond Polymer Refine Red Detection Kit (Leica Biosystems). Slides were then incubated with Protein Block X0909 (Dako/Agilent) for 30 min prior to application of primary antibodies (Iba1, FUJIFILM Wako, 019-19741, 1:800; wide-spectrum cytokeratin, Abcam ab9377, 1:100; Myeloperoxidase, Abcam ab9535, 1:30) for 1 h. Antibodies were conjugated to an immunofluorescence probe with Donkey anti-Rabbit IgG, Alexa Fluor 488, at 1:1000 (Invitrogen A21206). Nuclei were labeled by incubation with DAPI at 1:1000 (Thermo Fisher 62248). Sections were examined by immunofluorescence microscopy using an Olympus BX51 microscope and photomicrographs were taken using an Olympus DP73 camera.

**Infection of bone marrow-derived neutrophils and macrophages.** Mouse bone marrow neutrophils (BMDN) were isolated from total bone marrow cells by negative selection using magnetic bead-based EasySep Mouse Neutrophil Enrichment Kit (Stem Cell Technologies). Bone marrow-derived macrophage (BMDM) were cultured in MEM–F-12 supplemented with 0.1 IU/ml M-CSF, 10% heat-inactivated fetal bovine serum (Hyclone), 10 mM L-glutamine, and 1 μg/ml gentamicin. Both BMDN and BMDM were infected at an MOI of 40. The plates were rocked at 37 °C for 2 h. 1 mM ATP was added at 30 min prior to harvesting supernatants of CtD or CtD CT135$^-$ infected BMDN and BMDM for cytokine assay. Culture supernatants were assayed by ELISA for IL-1β secretion. LDH cytotoxicity was quantified by using CytoTox 96 Non-Radioactive Cytotoxicity Assay (Promega) according to the manufacturer's instructions. 1 mM ATP was added at 30 min prior to harvesting supernatants of CtD or CtD CT135$^-$ infected BMDN for LDH assay. To determine whether de novo protein synthesis was required for inflammasome activation, BMDN or BMDM were pretreated with rifampicin (10 μg/ml) prior to infection and post-infection. Anhydrotetracycline hydrochloride (aTC) (50 ng/ml) was added to BMDN prior-and post-infection for induction of CT135.

**Cell isolation and flow cytometry.** Mouse uterine horns were cut into small pieces, washed with HBSS containing 25 mM HEPES, 5 mM EDTA, and 10% heat-inactivated horse serum (Life Technologies) at room temperature for 15 min, and digested with RPMI 1640 containing 10% horse serum (Hyclone) with 1 mg/mL Collagenase D (Roche), 1 mg/mL Dispase II (Roche) and 0.25 mg/mL DNase I (Sigma) and shaken for 1 h at 37 °C. Cell suspensions were filtered through 70-μm nylon cell strainers. Cells were preincubated with anti-FcgRIII/II (Fc block) in FACS buffer for 30 min and stained for cell surface markers with fluorochrome-labeled antibodies. Cells were processed with the Becton Dickinson (BD) LSRII flow cytometer and All flow cytometry data are analyzed with FlowJo v10 software. The following antibodies were purchased from BioLegend: Brilliant Violet 785 anti-CD45 (clone 30-F11, 1:100); Brilliant Violet 421 PE/Cy7 anti-CD11b (clone M1/70, 1:100); 750 PE/Cy7 anti-F4/80 (clone BM8, 1:100); APC/Fire™ anti-Ly-6C (clone HK1.4, 1:100); FITC anti-I-A/I-E (clone M5/114.15.2, 1:100); FITC anti-CD206 (clone C068C2, 1:100); PerCP/Cyanine5.5 anti-CD80 (clone 16-10A1, 1:100); PE anti-IL-10 (clone JES5-16E3, 1:50). APC-TNF alpha (clone MP6-XT22, 1:50), APC anti-Arginase 1 (clone A1exF5, 1:50) and PE-iNOS (clone CXNFT, 1:50) were purchased from ThermoFisher Scientific.

**Immunofluorescence.** HeLa 229 cells (ATCC, catalog number CCL-2.1) were plated on coverslips in 24-well plates and infected with Ct at MOI = 0.3. 10 ng/ml of anhydrotetracycline hydrochloride (aTC) was added to induce the expression of CT135. The cells were fixed with cold methanol at eight hours after postinduction. Mouse anti-CT135 (1:100 dilution) and rabbit anti-IncA (1:100 dilution) were used for primary antibody staining. Secondary antibodies goat anti-mouse conjugated with Alexa Fluor 555 and goat anti-rabbit conjugated with Alexa Fluor 480 were used at 1:800. Coverslips were further stained with DAPI at 1:1000 in PBS for 5 min and mounted using Prolong Gold. Images were processed by Nikon Elements BR5.02.00 on a Nikon Eclipse 80i fluorescence microscope and analyzed by ImageJ Fiji.

**ATP measurement.** ATP release was quantified using an ATP assay kit (Sigma, MAK190). 100 μl supernatant from infected BM12.4 or BMDN cells were collected, immediately spun (1000 × g for 2 min, 4 °C), and analyzed via TECAN Spark.

**Proteomics.** HeLa 229 cells were plated in 6-well TC plates and were infected with Ct at MOI of 1. Cells were harvested in 1 ml hot 2% SDS-50 mM HEPES (pH 8.2) at 40 h pi and boiled for 10 min. Protein extracts from whole-cell lysates were trypsin digested, labeled by isobaric mass tags, and analyzed by shotgun proteomics using nano-scale liquid chromatographic tandem mass spectrometry on a Q-Exactive Plus mass spectrometer (Thermo Fisher). Differentially expressed

**a**

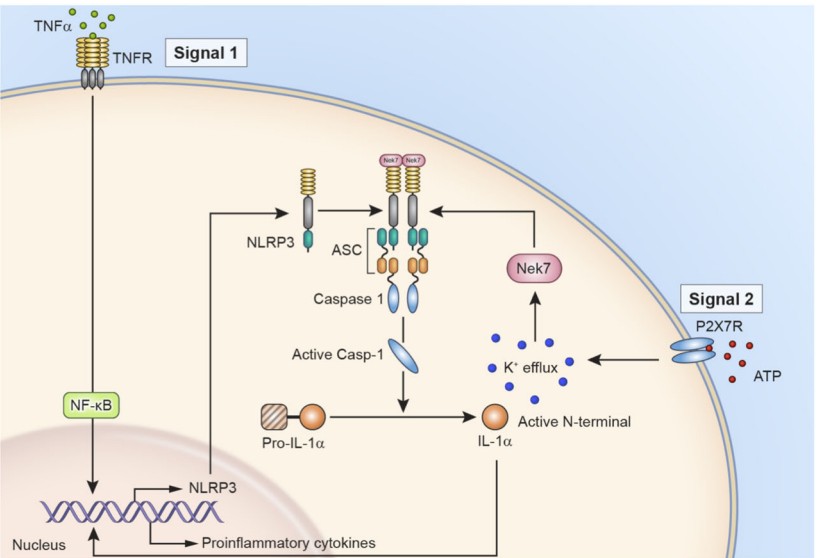

**b**

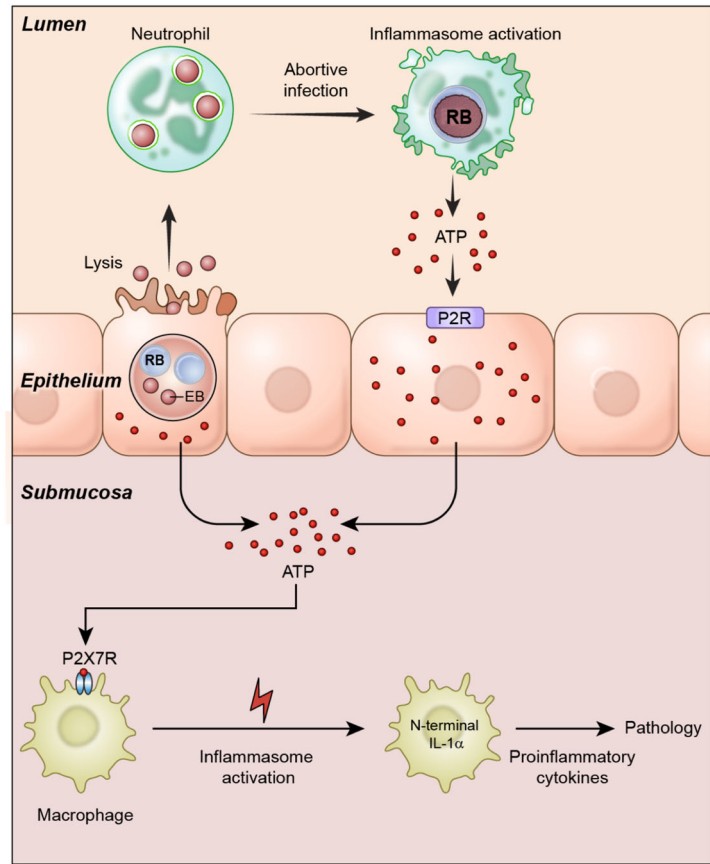

proteins were identified using the Linear Model for Micro Arrays (LIMMA) R package. The corresponding *P* values for each comparison were adjusted using the multiple testing procedure developed by Benjamini–Hochberg. Protein hits with adjusted *P* values of 0.05 and folds changes of >2 were considered statistically significant.

**Statistical analyses**. GraphPad Prism 7.0 software was used for data analysis. Statistical parameters including the exact value of *n*, (mean ± SD) and statistical significance is described in the figures. All the graphics used in the figures have been created by the authors. No images have been sourced from databases.

**Fig. 7 A model of sterile inflammation generated by chlamydial infection of uterine epithelium. a** NLRP3 inflammasome activation through ATP-P2X7 signaling pathway in phagocytic cells. **b** ATP released from chlamydiae infected oviduct epithelial cells (Fig. 6a) and luminal neutrophils function (i) directly through penetrating the lamina propria facilitated by matrix metallopeptidase 9 (MMP9) degradation of extracellular matrix molecules that has been shown to amplify the inflammatory response to chlamydial infection[56], or (ii) by paracrine/autocrine via P2R cascading purinergic signaling and exacerbation of the macrophage inflammatory response[57]. Because immunopathology occurs independently of the IL-1R and various PAMPs, we propose the processed N-terminal IL-1α is translocated to the nucleus to transcriptionally regulate pro-inflammatory cytokines which drive immunopathology. Notably, macrophage-associated immunopathology occurred independently of detectable chlamydial infection (Fig. 1i) presenting a model where mucosal infection generated DAMP signaling, which triggers sterile macrophage-associated inflammation in submucosal tissue.

**Reporting summary**. Further information on research design is available in the Nature Research Reporting Summary linked to this article.

## Data availability

The whole-genome sequence data of CtD and CtD CT135⁻ strains that support the findings of this study have been deposited with links to BioProject accession number PRJNA742023. Genome sequence data are available in GenBank or the NCBI (accession no. NC_017437, NC_017435.1). The data that support the findings of this study are available from the corresponding author upon reasonable request. Source data are provided with this paper.

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

## Acknowledgements

We thank Dr. Raymond Johnson from Yale University for providing BM12.4 cells. We thank Alan Hoofring and Erina He (Medical Arts Branch, NIH) for their assistance with graphic arts. This work was supported by the Intramural Research Program of the National Institute of Allergy and Infectious Diseases, National Institutes of Health.

## Author contributions

C.Y., L.L., J.W.M.C., M.B., L.M., G.L.S., H.S., A.K.K., D.D., C.B., K.W.B., I.N.M., C.-Y.C., C.A.M., and S.R. performed the experiments and analyzed data; M.Y. and K.T. provided GBP$^{\text{chr3}-/-}$ mice and Y.I. provided IL-1α$^{-/-}$ mice; C.Y., G.M., and H.D.C. wrote the paper. H.D.C. supervised the project.

## Competing interests

The authors declare no competing interests.
