## [Peer Review File · Nature Communications]

Chlamydia evasion of neutrophil host defense results in NLRP3 dependent myeloid-mediated sterile inflammation through the purinergic P2X7 receptorREVIEWER COMMENTS

Reviewer #1 (Remarks to the Author):

This manuscript describes a role of chlamydial CT135 in the activation of the NLRP3 inflammasome in neutrophils. The authors suggest that a signaling cascade is initiated by ATP released from neutrophils and also from epithelial cells that activates macrophages via the P2X7 receptor. Signaling via the P2X7 receptor then activates the macrophage NLRP3 inflammasome and thereby promotes inflammation.

The neutrophil specific effect of CT135 is somehow intriguing, however, several statements are not based on the results shown and are thus not justified.

General points:

The authors state that submucosal macrophage-mediated endometritis was driven by NLRP3 inflammasome activation through ATP-P2X7 signaling as a consequence of CT135 mediated neutrophil killing.

First, submucosal macrophage-mediated endometritis is not shown in the manuscript. Also, ATP-P2X7 signaling is neither investigated in macrophages, nor in any other cell. They instead used a P2X7 receptor knockout mouse, but many cells express P2X7R. NLRP3 inflammasome activation is already postulated to be mediated by TLR2-Myd88 and at least it is shown in vitro that there is no IL1b without TLR2/MyD88; but a link from ATP via P2X7R to NLRP3 is not investigated in any of the experiments shown. Finally, CT135-mediated neutrophil killing is not investigated in this manuscript. The main finding of this study therefore is that Chlamydia use CT135 to induce IL1b via TLR2 and the inflammasome. The other experiments in knockout mice are not well connected, and therefore the in vivo signalling cascade proposed here is not shown in the manuscript.

The CtD CT135- strain was generated by selection and has a point mutation in the CT135 gene. After this initial selection, the strain was sequenced and found to be isogenic except for the mutation in CT135. Irrespective of the sequencing information, the authors should use a CtD CT135- strain with complemented CT135 expression to confirm the role of CT135 in some crucial experiments. In the methods section of the manuscript the authors refer to such a complemented strain (CtD Ct135-:: bla CT135-) and it is not clear why they mention this but did not use it in this study.

It should be generally indicated in the figure legend, when post infection readouts were performed.

Major specific points:

- Figure 1a: The bacterial burden of wildtype and mutant infected mice differed only at day 3, the earliest timepoint investigated. Why did the authors start with day 3 and not earlier? There is already a drop of 2 orders of magnitude between the infection load and the bacterial burden of day 3, earlier timepoints may be significant as well.
- Figure 1g: Here the authors have to show the full gating strategy (live/dead, CD45, etc) and provide information on replicates and statistics. They should also provide the absolute numbers.
- In figure 1f and figure 1h, the IBA1 staining of mock versus CtD CT135- is inconsistent.
- In figure 3c, the authors use ILC- mice but it is not explained in the manuscript which mice exactly. They should clearly specify the mice they used and explain the evidence for the ILC deficiency.
- The results shown in figure 4d are not clear at all! The depletion of neutrophils in figure 3a affects only the mutant chlamydial strain but not the wildtype. In figure 4d, the wildtype is depleted 100 to 1000-fold compared to the tested knockouts which according to their model should be downstream of neutrophils. How can this be explained? Where both these experiments in figure 3a and 4d performed at day 7?
- In the experiments shown in figures 3b to f, bacterial load between chlamydial wildtype and mutant did not differ. Was there a difference in the KO mice tested in figure 4d?
- The concept shown in figure 4e includes TLR2 and MyD88 which according to the experiment shown in figure 4e have no significant impact on pathology. This is inconsistent.
- The authors state that NLRP3 inflammasome activation is essential for chronic infection and accompanying macrophage-mediated endometritis resulting in infertility. However, in Figure 5 NLRP3 knockout mice have the same number of bacteria after d7, suggesting that there is no difference in chronic infection. Furthermore, while gross pathology requires NLRP3, fertility has not been studied in NLRP3 knockout mice.

- The authors investigated the infiltration of macrophages in P2X7R^{-/-} and TNF- α p55^{-/-}/p75^{-/-} mice and conclude that “..the lack of gross pathology observed in KO mice (Fig. 6d-e) was not because macrophages failed to be recruited to the infection site.” These images show very different amounts of macrophages (see also point to Fig 1). Since no quantification was performed and replicates and statistics are missing, this conclusion is not justified. The authors should use FACS to quantify macrophages here.
- In figure 6g, the Chlamydia 16s staining is only seen in tissues of the wildtype mice, the knockout mice look exactly like non infected! These results do not fit to the quantification of vaginal swabs shown in figures 6c and d.

Reviewer #2 (Remarks to the Author):

In this manuscript Yang et al. investigate the innate immune response to Chlamydia infections in mice. Using a genital tract infection model they show that NLRP3 activation promotes chronic infection and infertility, and suggest that the Chlamydia CT135 protein drives NLRP3 activation via TLR2 as a pathogenic strategy. Finally, they propose a model in which macrophages sense Chlamydia-induced neutrophil death by detecting the ATP released by dying neutrophils via P2X7.

While the model which the authors propose is very interesting and would nicely integrate the interaction between macrophages, neutrophils and the epithelium, the experimental evidence does not support the model. Additional experiments will be necessary to 1) define how Chlamydia kills neutrophils, 2) show that the ATP which supposedly activate NLRP3 is selectively derived from neutrophils, 3) show that Chlamydia elicits the same inflammasome response in both macrophages and neutrophils, 4) that CT135 activates TLR3 and not NLRP3 directly, and 5) that macrophages NLRP3 activation in vivo is only driven by ATP and not Chlamydia infection.

Specific remarks:

Fig. 1g: provide same data for CtD CT135- infected mice

Fig. 2h: provide quantification

Fig 2f: It would be important to include data from CtD CT135- infected CCR2^{-/-} mice. If CT135 acts via CCR2, we would not expect any changes between CtD CT135- infected WT and CCR2^{-/-} mice.

Page 9, lines 169-171: Since other proteases, such as caspase-8 can also promote IL-1 β cleavage and release, as well as lytic cell death, the evidence at this point is not sufficient to implicate inflammasomes.

Page 10 lines 188-189: While CtD cause LDH release in both neutrophils and macrophages, the underlying signaling pathways might be different. Previous work has shown that canonical inflammasome activation has different outcome between neutrophils and macrophages. Thus BMDMs are not a viable substitute for neutrophils, and no conclusion can be drawn from BMDM experiments on the situation in neutrophils.

As the authors have the inflammasome KO mice available, they can generate BM-derived neutrophils easily

Lines 211-212: This conclusion needs to be corrected. The results show that TLR2 KO have reduced IL-1 β even upon CtD CT135 infection. This suggest that lack of CT125 results in reduced NLRP3 activation, but that the TLR2 signaling that primes NLRP3 is independent of CT135. Most likely it is CtD lipoproteins that drive TLR2 activation. A simple experiment would be to look at TLR2-dependent cytokine release in BMDMs infected with WT or CT135 mutant CtD.

Fig. 4e: This data support the importance of the NLRP3-Casp-1 axis, but do not confirm that this axis is active in neutrophils. Neutrophil-specific deletions would be required.

Fig. 6: It remains still unclear what pathway is activated in infected neutrophils to drive ATP release. Treatment with apyrase in vivo would further strengthen the findings. Finally, there is no evidence that P2X7 need to act in macrophages in vivo, and that ATP is derived from lysing neutrophils.

Reviewer #1 (Remarks to the Author):

This manuscript describes a role of chlamydial CT135 in the activation of the NLRP3 inflammasome in neutrophils. The authors suggest that a signaling cascade is initiated by ATP released from neutrophils and also from epithelial cells that activates macrophages via the P2X7 receptor. Signaling via the P2X7 receptor then activates the macrophage NLRP3 inflammasome and thereby promotes inflammation.

The neutrophil specific effect of CT135 is somehow intriguing, however, several statements are not based on the results shown and are thus not justified.

General points:

The authors state that submucosal macrophage-mediated endometritis was driven by NLRP3 inflammasome activation through ATP-P2X7 signaling as a consequence of CT135 mediated neutrophil killing.

First, submucosal macrophage-mediated endometritis is not shown in the manuscript.

We agree we have not directly demonstrated chlamydia infection causes macrophage-mediated endometritis. We have removed macrophage-mediated endometritis and replaced with macrophage associated pathology. We believe our revised manuscript support this conclusion.

Also, ATP-P2X7 signaling is neither investigated in macrophages, nor in any other cell. They instead used a P2X7 receptor knockout mouse, but many cells express P2X7R. NLRP3 inflammasome activation is already postulated to be mediated by TLR2-Myd88 and at least it is shown in vitro that there is no IL1 β without TLR2/MyD88; but a link from ATP via P2X7R to NLRP3 is not investigated in any of the experiments shown.

We have done the experiments suggested by the reviewer. BMDM derived from C57BL/6, NLRP3^{-/-}, Caspase1^{-/-} and P2X7R^{-/-} mice were incubated with the supernatant from CtD or CtD CT135⁻ infected neutrophils. IL-1 β secretion was determined by ELISA as shown in figure 6c. We included results in lines 267-274.

Finally, CT135-mediated neutrophil killing is not investigated in this manuscript.

CT135-mediated neutrophil killing experiments are shown in figure 3g-i based on LDH assay and IL-1 β secretion. We conducted additional experiments in BMDM derived from C57BL/6, TLR2^{-/-}, MyD88^{-/-}, NLRP3^{-/-}, Caspase1^{-/-} mice. These findings show IL-6 and IL-1 β secretion is CT135 dependent in figure 4a-c and result section (lines 195-205).

The main finding of this study therefore is that Chlamydia use CT135 to induce IL1b via TLR2 and the inflammasome. The other experiments in knockout mice are not well connected, and therefore the in vivo signaling cascade proposed here is not shown in the manuscript.

We believe the disconnect is because of chlamydia's complexity in interacting with different host cells in the model. (1). Chlamydiae are phagocytized by neutrophils. Chlamydiae undergo partial growth differentiation in neutrophils. During this infection process, CT135 triggers NLRP3 inflammasome activation via TLR2/MyD88 signaling as a pathogenic strategy to evade neutrophil host defense. (2). On the other hand, chlamydiae productively infect uterine horn epithelial cells likely CT135 functions similarly in activation via TLR2/MyD88 signaling, however NLRP3 is not expressed in uterine horn epithelial cells. Therefore, NLRP3 inflammasome is not activated. (3). In our model, submucosal macrophages are not infected by chlamydiae, the macrophage GT immunopathology is driven by ATP derived from infected neutrophils and epithelial cells that engages P2X7R mediated NLRP3 inflammasome activation.

We used NLCR4^{-/-}, AIM2^{-/-}, NOD1^{-/-}, and NOD2^{-/-} mice to confirm these inflammasome signaling pathways do not contribute to GT pathology in our model. We have made this connection clearer in lines 250-255.

The CtD CT135⁻ strain was generated by selection and has a point mutation in the CT135 gene. After this initial selection, the strain was sequenced and found to be isogenic except for the mutation in CT135. Irrespective of the sequencing information, the authors should use a CtD CT135⁻ strain with complemented CT135 expression to confirm the role of CT135 in some crucial experiments. In the methods section of the manuscript the authors refer to such a complemented strain (CtD Ct135^{-::} bla CT135⁻) and it is not clear why they mention this but did not use it in this study.

Allelic genetic exchange is not available for *Chlamydia trachomatis*. Therefore plasmid-based complementation is used. We made the complement strain CtD Ct135^{-::} bla CT135 for our studies. The CtD Ct135^{-::} bla CT135 is highly toxic following induction by tetracycline as shown in Extended Data Fig. 2. The toxicity is due to the high plasmid copy number in the transformants and elevated CT135 expression. Therefore, the plasmid complement strain could not be used in our studies. In lines 104-107 we added the sentence "Allelic genetic is not available in chlamydiae and tetracycline induced plasmid-based complementation of CtD Ct135^{-::} bla CT135 is highly toxic (Extended Data Fig. 2) due to the high plasmid copy number in the transformants¹ making in vitro and in vivo experiments impractical."

To emphasize the toxicity of complement strain CtD Ct135^{-::} bla CT135, we have added a one-step growth curve demonstrating the toxicity of the strain (Extended Data Fig. 2e).

It should be generally indicated in the figure legend, when post infection readouts were performed.

We have added times post infection in the figure legends.

Major specific points:

- Figure 1a: The bacterial burden of wildtype and mutant infected mice differed only at day 3, the earliest timepoint investigated. Why did the authors start with day 3 and not earlier? There is already a drop of 2 orders of magnitude between the infection load and the bacterial burden of day 3, earlier timepoints may be significant as well.

The chlamydial growth cycle modulates between noninfectious replicating reticular bodies and infectious elementary bodies, this cycle requires 48 to 72 hours for completion. Therefore, culture results from infected mice at days one and two are characteristically low compared to day 3 which is the peak of infectious loads.

- Figure 1g: Here the authors have to show the full gating strategy (live/dead, CD45, etc) and provide information on replicates and statistics. They should also provide the absolute numbers.

We repeated the flow cytometry experiments in the entire study. The information on full gating strategy, replicates, statistics and absolute numbers now is shown in the updated figures (Figure 1, Figure 2, Figure 6, Extended Data Fig. 3, Extended Data Fig. 4, Extended Data Fig. 5 and Extended Data Fig. 8) and results section (lines 117-127, 147-152, 285-297).

- In figure 1f and figure 1h, the IBA1 staining of mock versus CtD CT135- is inconsistent.

We agree that immunohistochemical staining is not sufficient to address quantitative differences among groups of mice. Therefore, we performed additional experiments to address this criticism. Total leukocytes were isolated from entire GT and CD11b⁺ F4/80⁺ macrophages were quantitated by flow cytometry. We found total CD11b⁺ F4/80⁺ macrophages were significant higher in CtD and CT135⁻ infected mice compared to mock infection but did not differ between the two groups of infected mice. We have modified figure 1 to include these new results (lines 117-127).

- In figure 3c, the authors use ILC⁻ mice but it is not explained in the manuscript which mice exactly. They should clearly specify the mice they used and explain the evidence for the ILC deficiency.

Rag2^{-/-} Il2rg^{-/-} (Ragyc^{-/-}) mice were purchased from Taconic Farms (CatLog number # 4111-F) which lack T and B cells and innate lymphoid cells (ILCs). The mouse strain has been used to study the role of ILC in innate immunity by other investigators^{2,3,4}. We have added mouse strain information in line 405-406 “deficient in T, B and innate lymphoid cells (ILCs)”.

- The results shown in figure 4d are not clear at all! The depletion of neutrophils in figure 3a affects only the mutant chlamydial strain but not the wildtype. In figure 4d, the wildtype is depleted 100 to 1000-fold compared to the tested knockouts which according to their model should be downstream of neutrophils. How can this be explained? Where both these experiments in figure 3a and 4d performed at day 7?

The mouse strains used in figure 3a and figure 4d are different. RAG2^{-/-} mice were used in figures 3a to f. C57BL/6, TLR2^{-/-}, MyD88^{-/-}, NLRP3^{-/-} and Casp1^{-/-} were used in figure 4d. CtD infected C57BL/6 and RAG2 exhibit differences in infectious burden at day 7 as shown in figures 1a and 2a. These results suggest adaptive immunity may function in reducing infectious loads in C57BL/6 mice at day 7.

We believe the disconnect is because of chlamydia's complexity in interacting with different host cells in the infection model. (1) Chlamydia are phagocytized by neutrophils. Chlamydiae undergo partial growth differentiation in neutrophils. Chlamydia CT135 triggers NLRP3 inflammasome activation via TLR2/MyD88 signaling as a pathogenic strategy to evade neutrophil host defense. (2) Chlamydia productively infect uterine horn epithelial cells. Since NLRP3 is not expressed in uterine horn epithelial cells⁵, NLRP3 inflammasome is not activated in infected uterine horn epithelial cells. However, TLR2/MyD88 signaling was activated in epithelial cells which is known to occur⁶. (3) Chlamydia infected dendritic cells activated Caspase1 and induce secretion of IL-1beta and IL-18⁷.

We agree we should not observe difference burden in TLR2^{-/-}, MyD88^{-/-}, NLRP3^{-/-} and Casp1^{-/-} mice based on we found CT135 dependent NLRP3 inflammasome in neutrophils. However, considering points (2) and (3) described above, these findings could explain the early higher burdens in TLR2^{-/-}, MyD88^{-/-}, NLRP3^{-/-} and Casp1^{-/-} mice.

Yes, day 7 was used for culturing chlamydia.

- In the experiments shown in figures 3b to f, bacterial load between chlamydial wildtype and mutant did not differ. Was there a difference in the KO mice tested in figure 4d?

Yes, they are different. RAG2^{-/-} mice were used for antibody treatment, depletion in figures 3b to f. C57BL/6, TLR2^{-/-}, MyD88^{-/-}, NLRP3^{-/-} and Casp1^{-/-} were used in figure 4d.

CtD infected RAG2^{-/-} mice were 10-fold higher than CtD CT135⁻ infected RAG2^{-/-} mice at day 7 post infection. We performed statistical analysis on these data and the difference between the two groups are significant. We have added this information to a modified figure 3a-3f.

- The concept shown in figure 4e includes TLR2 and MyD88 which according to the experiment shown in figure 4e have no significant impact on pathology. This is inconsistent.

This is a great question. We showed submucosal macrophages lack detectable chlamydial organisms (figure 1) however these mice develop macrophage associated GT pathology

mediated by NLRP3 inflammasome via P2X7R signaling pathway (figure 6). We proposed ATP released from infected neutrophils and oviduct cells could be the source for activating P2X7R signaling pathway responsible for GT pathology. The TLR2 and MyD88 signaling pathway is important for activating CtD CT135 dependent NLRP3/Caspase1 inflammasome in neutrophils. Infectious burden is significantly higher in CtD infected MyD88^{-/-} mice compared to NLRP3^{-/-} or Casp1^{-/-} mice suggesting MyD88 signaling pathway was activated in epithelial cells which is known to occur⁶. Therefore, the early higher burden in infected TLR2^{-/-} and MyD88^{-/-} uterine epithelial cells is major source of the extracellular ATP for activating P2X7R mediated NLRP3 inflammasome activation. Therefore, this could be the reason infected TLR2^{-/-} and MyD88^{-/-} mice develop pathology.

The authors state that NLRP3 inflammasome activation is essential for chronic infection and accompanying macrophage-mediated endometritis resulting in infertility. However, in Figure 5 NLRP3 knockout mice have the same number of bacteria after d7, suggesting that there is no difference in chronic infection. Furthermore, while gross pathology requires NLRP3, fertility has not been studied in NLRP3 knockout mice.

This is an interesting and important question. It is clear in our study there is strong correlation with early infectious loads and late GT immunopathology. We do not fully understand this, but we propose early high infection loads trigger NLRP3-dependent macrophage programming and imprinting important for development of late pathology. We included this interpretation in lines 232-240.

As suggested, we performed fertility study in NLRP3^{-/-} mice. CtD infected NLRP3^{-/-} mice were fertile. We included these results in figure 5b and added them into the results section (line 223).

- The authors investigated the infiltration of macrophages in P2X7R^{-/-} and TNF-α p55^{-/-}/p75^{-/-} mice and conclude that “the lack of gross pathology observed in KO mice (Fig. 6d-e) was not because macrophages failed to be recruited to the infection site.” These images show very different amounts of macrophages (see also point to Fig 1). Since no quantification was performed and replicates and statistics are missing, this conclusion is not justified. The authors should use FACS to quantify macrophages here.

We agree that immunohistochemical staining is not sufficient to address quantitative differences among groups of mice. Therefore, we performed additional experiments to address this criticism. Total leukocytes were isolated from entire GT and CD11b⁺ F4/80⁺ macrophages were quantitated by flow cytometry. We found total CD11b⁺ F4/80⁺ macrophages were significant higher in number in CtD infected C57BL/6, P2X7R^{-/-}, and TNF-α p55^{-/-}/p75^{-/-} compared to mock infection but did not differ among the groups. We have modified figure 6 and included these new results. We also modified the result sections according in lines 285-290.

- In figure 6g, the Chlamydia 16s staining is only seen in tissues of the wildtype mice, the knockout mice look exactly like non infected! These results do not fit to the quantification of vaginal swabs shown in figures 6c and d.

We agree Chlamydia 16s ISH staining does not correlated directly with chlamydia loads. Chlamydia 16s ISH staining was performed by veterinary pathologist in a blinded fashion, and infectious burdens are low at later infection period, therefore, the images were likely not as representative as the more quantitative recoverable IFU data. Therefore, we have removed the ISH staining and based our conclusion on recoverable IFU data.

Reviewer #2 (Remarks to the Author):

In this manuscript Yang et al. investigate the innate immune response to Chlamydia infections in mice. Using a genital tract infection model, they show that NLRP3 activation promotes chronic infection and infertility and suggest that the Chlamydia CT135 protein drives NLRP3 activation via TLR2 as a pathogenic strategy. Finally, they propose a model in which macrophages sense Chlamydia-induced neutrophil death by detecting the ATP released by dying neutrophils via P2X7.

We appreciate the reviewer' comments. We found them both helpful and constructive. We have done additional experiments to address reviewer' concerns.

1) define how Chlamydia kills neutrophils

CT135-mediated neutrophil killing experiments are shown in figure 3g-i based on LDH assay and IL-1 β secretion. We conducted additional experiments in BMDM derived from C57BL/6, TLR2^{-/-}, MyD88^{-/-}, NLRP3^{-/-}, Caspase1^{-/-} mice. These findings show IL-6 and IL-1 β secretion is CT135 dependent in figure 4a-c. A mechanism depicting how chlamydial CT135 dependent neutrophil killing occurs via TLR2/MyD88 signaling pathway resulting in NLRP3 activation and cell death.

2) show that the ATP which supposedly activate NLRP3 is selectively derived from neutrophils.

We have done the experiments suggested by the reviewer. BMDM derived from C57BL/6, NLRP3^{-/-}, Caspase1^{-/-} and P2X7R^{-/-} mice were incubated with the supernatant from CtD or CtD CT135⁻ infected neutrophils. IL-1 β secretion was determined by ELISA as shown in figure 6c. We included results in lines 267-274.

3) show that Chlamydia elicits the same inflammasome response in both macrophages and neutrophils

CT135 dependent NLRP3/Caspase1 inflammasome activation via TLR2 and MyD88 signaling pathway was performed in bone marrow derived neutrophils (BMDN) as shown in figure 4a and 4c. We included results in lines 195-205.

4) that CT135 activates TLR3 and not NLRP3 directly,

CT135 dependent cytokine release via TLR2 and MyD88 signaling pathway in BMDNs were done as shown in figure 4a. We included results in lines 201-203.

5) that macrophages nLRP3 activation in vivo is only driven by ATP and not Chlamydia infection.

By our ISH staining of infected tissues, our veterinary pathologist, did not detect chlamydiae in submucosal tissues containing infiltrating macrophages, despite careful examination of intact genital tracts. Based on this finding, we conclude ATP derived from uterine epithelial cells and luminal neutrophil cells are the primary source for activating submucosal macrophage NLRP3 inflammasome.

Specific remarks:

Fig. 1g: provide same data for CtD CT135- infected mice

We repeated the experiments, performed flow cytometry, and included CtD CT135- infected mice in figure (1f-h and Extended Data Fig. 4). We also added this information to the result section (lines 116-127).

Fig. 2h: provide quantification

Total leukocytes were isolated from the entire GTs and CD11b⁺ F4/80⁺ macrophages were quantitated by flow cytometry. We found total CD11b⁺ F4/80⁺ macrophages were significant higher in CtD and CT135- infected mice compared to mock infection but did not differ between the two groups of infected mice. We have modified figure (2f-h and Extended Data Fig. 5) to include these new results. We also added this information to the result section (lines 147-152).

Fig 2f: It would be important to include data from CtD CT135- infected CCR2^{-/-} mice. If CT135 acts via CCR2, we would not expect any changes between CtD CT135- infected WT and CCR2^{-/-} mice.

We infected CCR2^{-/-} mice with CtD CT135- strain. We observed higher infectious burdens in CCR2^{-/-} mice infected with CtD CT135- at day 3 post infection but not at later time points pi. We did not observe pathology difference in CtD CT135- infected C57BL/6 and CCR2^{-/-} mice. We added these results in figure (2i-j) and included these findings in the results section (lines 156-159).

Page 9, lines 169-171: Since other proteases, such as caspase-8 can also promote IL-1b cleavage and release, as well as lytic cell death, the evidence at this point is not sufficient to implicate inflammasomes.

We very much appreciate this comment. We have modified the sentence in lines 191-192 “CT135 functions as an inclusion membrane pore protein that secretes PAMP(s) into the neutrophil cytosol resulting in IL-1 β secretion and lytic cell death.”

Page 10 lines 188-189: While CtD cause LDH release in both neutrophils and macrophages, the underlying signaling pathways might be different. Previous work has shown that canonical inflammasome activation has different outcome between neutrophils and macrophages. Thus, BMDMs are not a viable substitute for neutrophils, and no conclusion can be drawn from BMDM experiments on the situation in neutrophils. As the authors have the inflammasome KO mice available, they can generate BM-derived neutrophils easily

We have moved BMDMs results to supplement section (Extended Data Fig. 6) and done additional experiments using bone marrow derived neutrophils (BMDN). Ideally, it would be preferable to include all the inflammasome KO mice in these new experiments. However, due to the COVID 19 pandemic our animal facility and staff were restricted limiting our mouse usage. Therefore, we have performed new experiments using BMDN in selected knockout mice that showed a phenotype in BMDM. We included results in lines 195-205.

Lines 211-212: This conclusion needs to be corrected. The results show that TLR2 KO have reduced IL-1b even upon CtD CT135 infection. This suggest that lack of CT125 results in reduced NLRP3 activation, but that the TLR2 signaling that primes NLRP3 is independent of CT135. Most likely it is CtD lipoproteins that drive TLR2 activation. A simple experiment would be to look at TLR2-dependent cytokine release in BMDMs infected with WT or CT135 mutant CtD.

As suggested, we did TLR2-dependent cytokine release in BMDNs infected with WT or CT135 mutant CtD. The results are included in Figure 4a and results section (lines 201-203)

Fig. 4e: This data support the importance of the NLRP3-Casp-1 axis, but do not confirm that this axis is active in neutrophils. Neutrophil-specific deletions would be required.

We performed new experiments using infected bone marrow derived neutrophils (BMDN) from C57BL/6, NLRP3^{-/-}, Casp1^{-/-}. BMDNs were infected with CtD and CtD CT135⁻ and IL-6 or IL-1 β in supernatants was measured by ELISA. These results are included in figure (4a-b) and results section (lines 201-205).

Antibody specific depletion of neutrophils is only practical for a maximal period 2-3 weeks⁸. In our model GT pathology does not develop until 60 days post infection. We think this short time period of treatment may not be sufficient to show pathology difference in CtD infected NLRP3^{-/-}, Casp1^{-/-} shown in figure 4e.

Fig. 6: It remains still unclear what pathway is activated in infected neutrophils to drive ATP release. Treatment with apyrase in vivo would further strengthen the findings. Finally, there is no

evidence that P2X7 needs to act in macrophages *in vivo*, and that ATP is derived from lysing neutrophils.

As suggested, we treated mice with apyrase (15 IU/mice)⁹. Mice were pretreated, infected, and treated daily for one-week post infection. We monitored the infectious burden weekly and GT pathology at day 60 pi. We did not observe an effect on infection of GT pathology at 60 days PI. Since ATP is produced by numerous cell types in multiple tissues and likely enriched in chlamydial infected mucosal tissues, apyrase treatment may not be sufficient enough to exhibit biological phenotype. Secondly, it is possible that continuous apyrase treatment over the entire course of infection (60 days) is necessary to alter macrophage-associated late pathology. Such a prolonged treatment regime was not attempted as it was unrealistic due to both cost and likely deleterious effect it would have on the general health of mice.

Fig. 1. Apyrase treatment of CtD infected C57BL/6.

a. Infection kinetics. b. Quantification of gross uterine horn pathology at day 56 pi. c. Flow cytometry of total CD11b⁺ F4/80⁺ macrophages.

Fig. 2 Flow cytometry of CD11b⁺ F4/80⁺ Ly6C⁺ macrophages in entire GT.

a, Ly6C⁺ MHC II⁺ macrophages. b, Ly6C⁺ iNOS⁺ macrophages. c, Ly6C⁺ CD206⁺ macrophages. d, Ly6C⁺ Arginase⁺ macrophages. e, Ly6C⁺ TNF- α ⁺ macrophages. f, Ly6C⁺ IL-10⁺ macrophages. (NS, not significant, *p <0.05, **p <0.01, student's t tests)

References

1. Wang Y, Kahane S, Cutcliffe LT, Skilton RJ, Lambden PR, Clarke IN. Development of a transformation system for *Chlamydia trachomatis*: restoration of glycogen biosynthesis by acquisition of a plasmid shuttle vector. *PLoS Pathog* **7**, e1002258 (2011).
2. Huang Y, *et al.* IL-25-responsive, lineage-negative KLRG1(hi) cells are multipotential 'inflammatory' type 2 innate lymphoid cells. *Nat Immunol* **16**, 161-169 (2015).
3. Abt MC, *et al.* Innate Immune Defenses Mediated by Two ILC Subsets Are Critical for Protection against Acute *Clostridium difficile* Infection. *Cell Host Microbe* **18**, 27-37 (2015).
4. Bando JK, Colonna M. Innate lymphoid cell function in the context of adaptive immunity. *Nat Immunol* **17**, 783-789 (2016).
5. Kummer JA, *et al.* Inflammasome components NALP 1 and 3 show distinct but separate expression profiles in human tissues suggesting a site-specific role in the inflammatory response. *J Histochem Cytochem* **55**, 443-452 (2007).
6. O'Connell CM, Ionova IA, Quayle AJ, Visintin A, Ingalls RR. Localization of TLR2 and MyD88 to *Chlamydia trachomatis* inclusions. Evidence for signaling by intracellular TLR2 during infection with an obligate intracellular pathogen. *The Journal of biological chemistry* **281**, 1652-1659 (2006).
7. Gervassi A, Alderson MR, Suchland R, Maisonneuve JF, Grabstein KH, Probst P. Differential regulation of inflammatory cytokine secretion by human dendritic cells upon *Chlamydia trachomatis* infection. *Infect Immun* **72**, 7231-7239 (2004).
8. Boivin G, *et al.* Durable and controlled depletion of neutrophils in mice. *Nat Commun* **11**, 2762 (2020).
9. Cauwels A, Rogge E, Vandendriessche B, Shiva S, Brouckaert P. Extracellular ATP drives systemic inflammation, tissue damage and mortality. *Cell Death Dis* **5**, e1102 (2014).

REVIEWER COMMENTS

Reviewer #1 (Remarks to the Author):

The authors significantly improved the manuscript. Some issues remain.

Previous comment:

The CtD CT135- strain was generated by selection and has a point mutation in the CT135 gene. After this initial selection, the strain was sequenced and found to be isogenic except for the mutation in CT135. Irrespective of the sequencing information, the authors should use a CtD CT135- strain with complemented CT135 expression to confirm the role of CT135 in some crucial experiments. In the methods section of the manuscript the authors refer to such a complemented strain (CtD Ct135-:: bla CT135-) and it is not clear why they mention this but did not use it in this study.

This is an important point since the entire study is based on a single chlamydial mutant. The explanation they provided why they did not use the complemented strain is not satisfying. The results they show rather demonstrate that tetracycline has a toxic effect on Chlamydia, no evidence for any overexpression of CT135 is provided (controls are missing). It is well known that Chlamydia is very sensitive to tetracycline treatment, and bit less to anhydrotetracycline which may be an alternative to induce the expression. In any case they have to show in a few critical experiments that complementing CT135 reverts the effect of the mutant or they have to provide the sequence of the mutant and show that it has only the mutation in CT135.

Reviewer #2 (Remarks to the Author):

The authors have addressed my comments with additional experiments. I have no further suggestions.

Reviewer #1 (Remarks to the Author):

The authors significantly improved the manuscript. Some issues remain.

Previous comment:

The CtD CT135⁻ strain was generated by selection and has a point mutation in the CT135 gene. After this initial selection, the strain was sequenced and found to be isogenic except for the mutation in CT135. Irrespective of the sequencing information, the authors should use a CtD CT135⁻ strain with complemented CT135 expression to confirm the role of CT135 in some crucial experiments. In the methods section of the manuscript the authors refer to such a complemented strain (CtD Ct135⁻:: bla CT135⁻) and it is not clear why they mention this but did not use it in this study.

This is an important point since the entire study is based on a single chlamydial mutant. The explanation they provided why they did not use the complemented strain is not satisfying. The results they show rather demonstrate that tetracycline has a toxic effect on Chlamydia, no evidence for any overexpression of CT135 is provided (controls are missing). It is well known that Chlamydia is very sensitive to tetracycline treatment, and bit less to anhydrotetracycline which may be an alternative to induce the expression. In any case they have to show in a few critical experiments that complementing CT135 reverts the effect of the mutant or they have to provide the sequence of the mutant and show that it has only the mutation in CT135.

We appreciate the constructive comments.

We included the whole genome sequence of wild type CtD and CtD CT135⁻ mutant strains. The CtD CT135⁻ strain is isogenic to CtD with the exception of a single base transition (C>T) at nucleotide position 152743 leading to an Arg201 (CGA) change introducing a stop codon (TGA). This mutation centrally disrupts the CT135 open reading frame. The data have been deposited with links to BioProject accession number PRJNA742023 in the NCBI BioProject database (<https://www.ncbi.nlm.nih.gov/bioproject/>). We have included these results in the revised manuscript (lines 404-413 and 424-432).

We performed complementation studies using the CtD CT135⁻ mutant strain. Both cytotoxicity and IL-1 β secretion, two CT135 dependent phenotypes, were restored when the CtD CT135⁻:: bla CT135 strain was used to infect BMDN (Figure 3h and 3i). In addition, a marked increase of IL-1 β was found when supernatants from CtD CT135⁻::bla CT135 strain infected BMDNs was added to cultured uninfected BMDMs (Figure 6c). We have modified Figure 3 and Figure 6 in the revised manuscript to include these new results (lines 185-187 and 279-281).

Reviewer #2 (Remarks to the Author):

The authors have addressed my comments with additional experiments. I have no further suggestions.

We appreciate the constructive comments during the review which have substantially improved the manuscript.